# Spec-Gaussian: Anisotropic View-Dependent Appearance for 3D Gaussian Splatting

**Ziyi Yang**[1,3]    **Xinyu Gao**[1]    **Yang-Tian Sun**[2]    **Yi-Hua Huang**[2]    **Xiaoyang Lyu**[2]
**Wen Zhou**[3]    **Shaohui Jiao**[3]    **Xiaojuan Qi**[2†]    **Xiaogang Jin**[1†]

[1]State Key Lab of CAD&CG, Zhejiang University
[2]The University of Hong Kong    [3]ByteDance Inc.

## Abstract

The recent advancements in 3D Gaussian splatting (3D-GS) have not only facilitated real-time rendering through modern GPU rasterization pipelines but have also attained state-of-the-art rendering quality. Nevertheless, despite its exceptional rendering quality and performance on standard datasets, 3D-GS frequently encounters difficulties in accurately modeling specular and anisotropic components. This issue stems from the limited ability of spherical harmonics (SH) to represent high-frequency information. To overcome this challenge, we introduce *Spec-Gaussian*, an approach that utilizes an anisotropic spherical Gaussian (ASG) appearance field instead of SH for modeling the view-dependent appearance of each 3D Gaussian. Additionally, we have developed a coarse-to-fine training strategy to improve learning efficiency and eliminate floaters caused by overfitting in real-world scenes. Our experimental results demonstrate that our method surpasses existing approaches in terms of rendering quality. Thanks to ASG, we have significantly improved the ability of 3D-GS to model scenes with specular and anisotropic components without increasing the number of 3D Gaussians. This improvement extends the applicability of 3D GS to handle intricate scenarios with specular and anisotropic surfaces. Our codes and datasets are available at https://ingra14m.github.io/Spec-Gaussian-website.

## 1   Introduction

High-quality reconstruction and photorealistic rendering from a collection of images are crucial for a variety of applications, such as augmented reality/virtual reality (AR/VR), 3D content production, and art creation. Classic methods employ primitive representations, like meshes [39] and points [4, 67], and take advantage of the rasterization pipeline optimized for contemporary GPUs to achieve real-time rendering. In contrast, neural radiance fields (NeRF) [37, 5, 38] utilize neural implicit representation to offer a continuous scene representation and employ volumetric rendering to produce rendering results. This approach allows for enhanced preservation of scene details and more effective reconstruction of scene geometries.

Recently, 3D Gaussian Splatting (3D-GS) [23] has emerged as a leading technique, delivering state-of-the-art quality and real-time speed. This method optimizes a set of 3D Gaussians that capture the appearance and geometry of a 3D scene simultaneously, offering a continuous representation that preserves details and produces high-quality results. Besides, the CUDA-customized differentiable rasterization pipeline for 3D Gaussians enables real-time rendering even at high resolution.

---

† Corresponding Authors.

38th Conference on Neural Information Processing Systems (NeurIPS 2024).

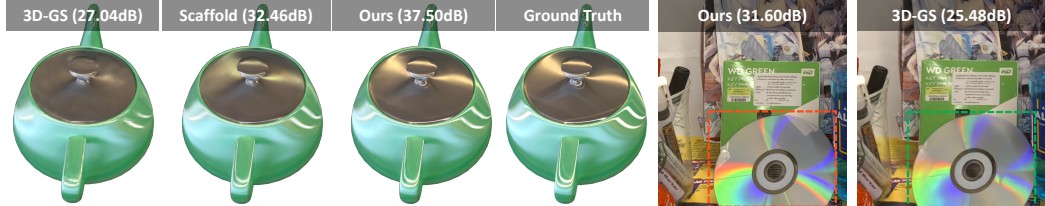

Figure 1: Our method not only achieves real-time rendering but also significantly enhances the capability of 3D-GS to model scenes with specular and anisotropic components. Key to this enhanced performance is our use of ASG appearance field to model the appearance of each 3D Gaussian, which results in substantial improvements in rendering quality for both complex and general scenes.

Despite its exceptional performance, 3D-GS struggles to model specular components within scenes (see Fig. 1). This issue primarily stems from the limited ability of low-order spherical harmonics (SH) to capture the high-frequency information in these scenarios. Consequently, this poses a challenge for 3D-GS to model scenes with reflections and specular components, as illustrated in Fig. 1.

To address the issue, we introduce a novel approach called *Spec-Gaussian*, which combines anisotropic spherical Gaussian (ASG) [60] for modeling anisotropic and specular components, an effective training mechanism to eliminate floaters and improve learning efficiencies, and anchor-based 3D Gaussians for acceleration and storage reduction. Specifically, the method incorporates two key designs: 1) A new 3D Gaussian representation that utilizes an ASG appearance field instead of SH to model the appearance of each 3D Gaussian. ASG with a few orders can effectively model high-frequency information that low-order SH cannot. This new design enables 3D-GS to more effectively model anisotropic and specular components in static scenes. 2) A coarse-to-fine training scheme specifically tailored for 3D-GS is designed to eliminate floaters and boost learning efficiency. This strategy effectively shortens learning time by optimizing low-resolution rendering in the initial stage, preventing the need to increase the number of 3D Gaussians and regularizing the learning process to avoid the generation of unnecessary geometric structures that lead to floaters.

By combining these advances, our approach can render high-quality results for specular highlights and anisotropy as shown in Fig. 4 while preserving the efficiency of Gaussians. Furthermore, comprehensive experiments reveal that our method not only endows 3D-GS with the ability to model specular highlights but also achieves state-of-the-art results in general benchmarks.

In summary, the major contributions of our work are as follows:

- A novel ASG appearance field to model the view-dependent appearance of each 3D Gaussian, which enables 3D-GS to effectively represent scenes with specular and anisotropic components.
- A coarse-to-fine training scheme that effectively regularizes training to eliminate floaters and improve the learning efficiency of 3D-GS in real-world scenes.
- An anisotropic dataset has also been made to assess the capability of our model in representing anisotropy.

## 2 Related Work

### 2.1 Implicit Neural Radiance Fields

Neural rendering has attracted significant interest in the academic community for its unparalleled ability to generate photorealistic images. Methods like NeRF [37] utilize Multi-Layer Perceptrons (MLPs) to model the geometry and radiance fields of a scene. Leveraging the volumetric rendering equation and the inherent continuity and smoothness of MLPs, NeRF achieves high-quality scene reconstruction from a set of posed images, establishing itself as the state-of-the-art (SOTA) method for novel view synthesis. Subsequent research has extended the utility of NeRF to various applications, including mesh reconstruction [53, 27, 58, 34], inverse rendering [48, 72, 31, 62], optimization of camera parameters [29, 55, 54, 41], few-shot learning [12, 61, 57], and anti-aliasing [2, 1, 3].

However, this stream of methods relies on ray casting rather than rasterization to determine the color of each pixel. Consequently, every sampling point along the ray necessitates querying the MLPs,

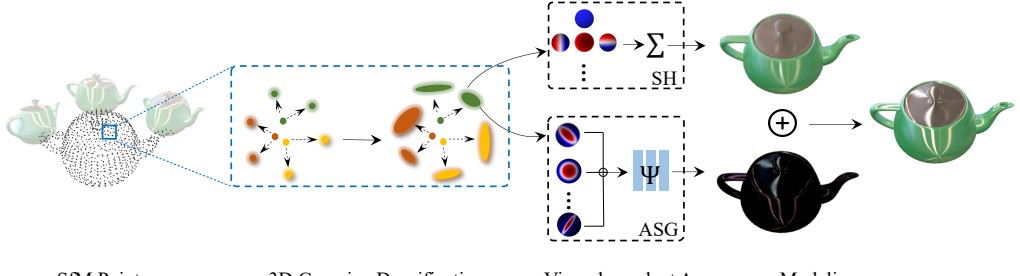

SfM Points          3D Gaussian Densification          View-dependent Appearance Modeling

Figure 2: **Pipeline of Spec-Gaussian.** The optimization process begins with SfM points derived from COLMAP or generated randomly, serving as the initial state for the 3D Gaussians. To address the limitations of low-order SH and pure MLP in modeling high-frequency information, we additionally employ ASG in conjunction with a feature decoupling MLP to model the view-dependent appearance of each 3D Gaussian. Then, 3D Gaussians with opacity $\sigma > 0$ are rendered through a differentiable Gaussian rasterization pipeline, effectively capturing specular highlights and anisotropy in the scene.

leading to significantly slow rendering speed and prolonged training convergence. This limitation substantially impedes their application in large-scene modeling and real-time rendering.

To reduce the training time of MLP-based NeRF methods and improve rendering speed, subsequent work has enhanced NeRF's efficiency in various ways. Structure-based techniques [68, 14, 43, 17, 7] have sought to improve inference or training efficiency by caching or distilling the implicit neural representation into more efficient data structures. Hybrid methods [30, 49] increase efficiency by incorporating explicit voxel-based data structures. Factorization methods [5, 18, 8, 16] apply a low-rank tensor assumption to decompose the scene into low-dimensional planes or vectors, achieving better geometric consistency. Compared to continuous implicit representations, the convergence of individual voxels in the grid is independent, significantly reducing training time. Additionally, Instant-NGP [38] utilizes a hash grid with a corresponding CUDA implementation for faster feature querying, enabling rapid training and interactive rendering of neural radiance fields. Spec-NeRF [35] achieves high-quality specular reflection modeling by introducing Gaussian directional encoding.

Despite achieving higher quality and faster rendering, these methods have not fundamentally overcome the substantial query overhead associated with ray casting. As a result, a notable gap remains before achieving real-time rendering. In this work, we build upon the recent 3D-GS [23], a point-based rendering method that leverages rasterization. Compared to ray casting-based methods, it significantly enhances both training and rendering speed.

## 2.2   Point-based Neural Radiance Fields

Point-based representations, similar to triangle mesh-based methods, can exploit the highly efficient rasterization pipeline of modern GPUs to achieve real-time rendering. Although these methods offer breakneck rendering speeds and are well-suited for editing tasks, they often suffer from holes and outliers, leading to artifacts in the rendered images. This issue arises from the discrete nature of point clouds, which can create gaps in the primitives and, consequently, in the rendered image.

To address these discontinuity issues, differentiable point-based rendering [67, 15, 24, 25] has been extensively explored for fitting complex geometric shapes. Notably, Zhang et al. [71] employ differentiable surface splatting and utilize a radial basis function (RBF) kernel to compute the contribution of each point to each pixel.

Recently, 3D-GS [23] has employed anisotropic 3D Gaussians, initialized from Structure from Motion (SfM), to represent 3D scenes. The innovative densification mechanism and CUDA-customized differentiable Gaussian rasterization pipeline of 3D-GS have not only achieved state-of-the-art (SOTA) rendering quality but also significantly surpassed the threshold of real-time rendering. Many concurrent works have rapidly extended 3D-GS to a variety of downstream applications, including dynamic scenes [33, 63, 64, 20, 26, 50], text-to-3D generation [28, 51, 9, 66, 10], avatars [74, 73, 21, 45, 40], scene editing [59, 6, 13], quality enhancement [36, 44] and mesh reconstruction [19, 11, 69, 34].

Despite achieving SOTA results on commonly used benchmark datasets, 3D-GS still struggles to model scenes with specular and reflective components, which limits its practical application in real-time rendering at the photorealistic level. In this work, by replacing spherical harmonics (SH) with an anisotropic spherical Gaussian (ASG) appearance field, we have enabled 3D-GS to model complex specular scenes more effectively.

# 3  Method

The overview of our method is illustrated in Fig. 2. The input to our model is a set of posed images of a static scene, together with a sparse point cloud obtained from SfM [46]. The core of our method is to use the ASG appearance field to replace SH in modeling the appearance of 3D Gaussians (Sec. 3.2). Moreover, we introduce a simple yet effective coarse-to-fine training strategy to reduce floaters in real-world scenes (Sec. 3.3). To further reduce the storage overhead and rendering speed pressure introduced by ASG, we combine a hybrid Gaussian model that employs sparse anchor Gaussians to facilitate the generation of neural Gaussians (Sec. 3.4) to model the 3D scene.

## 3.1  Preliminaries

**3D Gaussian splatting.**  3D-GS [23] is a point-based method that employs anisotropic 3D Gaussians to represent scenes. Each 3D Gaussian is defined by a center position $x$, opacity $\sigma$, and a 3D covariance matrix $\Sigma$, which is decomposed into a quaternion $r$ and scaling $s$. The view-dependent appearance of each 3D Gaussian is represented using the first three orders of spherical harmonics (SH). This method not only retains the rendering details offered by volumetric rendering but also achieves real-time rendering through a CUDA-customized differentiable Gaussian rasterization process. Following [75], the 3D Gaussians can be projected to 2D using the 2D covariance matrix $\Sigma'$, defined as:

$$\Sigma' = JV\Sigma V^T J^T, \tag{1}$$

where $J$ is the Jacobian of the affine approximation of the projective transformation, and $V$ represents the view matrix, transitioning from world to camera coordinates. To facilitate learning, the 3D covariance matrix $\Sigma$ is decomposed into two learnable components: the quaternion $r$, representing rotation, and the 3D-vector $s$, representing scaling. The resulting $\Sigma$ is thus represented as the combination of a rotation matrix $R$ and scaling matrix $S$ as:

$$\Sigma = RSS^T R^T. \tag{2}$$

The color of each pixel on the image plane is then rendered through a point-based volumetric rendering (alpha blending) technique:

$$C(\mathbf{p}) = \sum_{i \in N} T_i \alpha_i c_i, \quad \alpha_i = \sigma_i e^{-\frac{1}{2}(\mathbf{p}-\mu_i)^T \Sigma^{-1}(\mathbf{p}-\mu_i)}, \tag{3}$$

where $\mathbf{p}$ denotes the pixel coordinate, $T_i$ is the transmittance defined by $\Pi_{j=1}^{i-1}(1-\alpha_j)$, $c_i$ signifies the color of the sorted Gaussians associated with the queried pixel, and $\mu_i$ represents the coordinates of the 3D Gaussians when projected onto the 2D image plane.

**Anisotropic spherical Gaussian.**  Anisotropic spherical Gaussian (ASG) [60] has been designed in the traditional rendering pipeline to efficiently approximate lighting and shading. Different from spherical Gaussian (SG), ASG has been demonstrated to effectively represent anisotropic scenes with a small number. In addition to retaining the fundamental properties of SG, ASG also exhibits rotational invariance and can represent full-frequency signals. The ASG function is defined as:

$$ASG(\nu \mid [\mathbf{x}, \mathbf{y}, \mathbf{z}], [\lambda, \mu], \xi) = \xi \cdot S(\nu; \mathbf{z}) \cdot e^{-\lambda(\nu \cdot \mathbf{x})^2 - \mu(\nu \cdot \mathbf{y})^2}, \tag{4}$$

where $\nu$ is the unit direction serving as the function input; $\mathbf{x}$, $\mathbf{y}$, and $\mathbf{z}$ correspond to the tangent, bi-tangent, and lobe axis, respectively, and are mutually orthogonal; $\lambda \in \mathbb{R}^1$ and $\mu \in \mathbb{R}^1$ are the sharpness parameters for the $\mathbf{x}$- and $\mathbf{y}$-axis, satisfying $\lambda, \mu > 0$; $\xi \in \mathbb{R}^2$ is the lobe amplitude; S is the smooth term defined as $S(\nu; \mathbf{z}) = \max(\nu \cdot \mathbf{z}, 0)$.

Inspired by the power of ASG in modeling scenes with complex anisotropy, we propose integrating ASG into Gaussian splatting to join the forces of classic models with new rendering pipelines for

higher quality. For $N$ ASGs, we predefined orthonormal axes $\mathbf{x}$, $\mathbf{y}$, and $\mathbf{z}$, initializing them to be uniformly distributed across a hemisphere. During training, we allow the remaining ASG parameters, $\lambda$, $\mu$, and $\xi$, to be learnable. We use the reflect direction $\omega_r$ as the input to query ASG for modeling the view-dependent specular information. Note that we use $N = 32$ ASGs for each 3D Gaussian.

**Anchor-based Gaussian splatting.** Anchor-based Gaussian splatting was first proposed by Scaffold-GS [32]. Unlike the attributes carried by each entity in 3D-GS, each anchor Gaussian carries a position coordinate $\mathbf{P}_v \in \mathbb{R}^3$, a local feature $\mathbf{f}_v \in \mathbb{R}^{32}$, a displacement factor $\eta_v \in \mathbb{R}^3$, and $k$ learnable offsets $\mathbf{O}_v \in \mathbb{R}^{k \times 3}$. They use the COLMAP [46] point cloud to initialize each anchor 3D Gaussian, serving as the voxel centers to guide the generation of neural Gaussians. The position $\mathbf{P}_v$ of the anchor Gaussian is initialized as:

$$\mathbf{P}_v = \left\{ \left\lfloor \frac{\mathbf{P}}{\epsilon} + 0.5 \right\rfloor \right\} \cdot \epsilon, \tag{5}$$

where $\mathbf{P}$ is the point cloud position, $\epsilon$ is the voxel size, and $\{\cdot\}$ denotes removing duplicated anchors.

Then anchor Gaussians can guide the generation of neural Gaussians, which have the same attributes as vanilla 3D-GS. For each visible anchor Gaussian within the viewing frustum, we spawn $k$ neural Gaussians and predict their attributes. The positions $\mathbf{x}$ of neural Gaussians are calculated as:

$$\{\mathbf{x}_0, \ldots, \mathbf{x}_{k-1}\} = \mathbf{P}_v + \{\mathbf{O}_0, \ldots, \mathbf{O}_{k-1}\} \cdot \eta_v, \tag{6}$$

where $\mathbf{P}_v$ represents the position of the anchor Gaussian corresponding to $k$ neural Gaussians. The opacity $\sigma$ is calculated through a tiny MLP:

$$\{\sigma_0, \ldots, \sigma_{k-1}\} = \mathcal{F}_\sigma \left( \mathbf{f}_v, \delta_{cv}, \mathbf{d}_{cv} \right), \tag{7}$$

where $\delta_{cv}$ denotes the distance between the anchor Gaussian and the camera, and $\mathbf{d}_{cv}$ is the unit direction pointing from the camera to the anchor Gaussian. The rotation $r$ and scaling $s$ of each neural Gaussian are derived similarly using the corresponding tiny MLP $\mathcal{F}_r$ and $\mathcal{F}_s$.

## 3.2 Anisotropic View-Dependent Appearance

**ASG appearance field for 3D Gaussians.** Although SH has enabled view-dependent scene modeling, the low frequency of low-order SH makes it challenging to model scenes with complex optical phenomena such as specular highlights and anisotropic effects. Therefore, instead of using SH, we propose using an ASG appearance field based on Eq. (4) to model the appearance of each 3D Gaussian. However, the introduction of ASG increases the feature dimensions of each 3D Gaussian, raising the model's storage overhead. To address this, we employ a compact learnable MLP $\Theta$ to predict the parameters for $N$ ASGs, with each Gaussian carrying only additional local features $\mathbf{f} \in \mathbb{R}^{24}$ as the input to the MLP:

$$\Theta(\mathbf{f}) \rightarrow \{\lambda, \mu, \xi\}_N. \tag{8}$$

To better differentiate between high and low-frequency information and further assist ASG in fitting high-frequency specular details, we decompose color $c$ into diffuse and specular components:

$$c = c_d + c_s, \tag{9}$$

where $c_d$ represents the diffuse color, modeled using the first three orders of SH, and $c_s$ is the specular color calculated through ASG. We refer to this comprehensive approach to appearance modeling as the ASG appearance field.

Although ASG theoretically enhance the ability of SH to model anisotropy, directly using ASG to represent the specular color of each 3D Gaussian still falls short in accurately modeling anisotropic and specular components, as demonstrated in Fig. 6. Inspired by [16], we do not use ASG directly to represent color but instead employ ASG to model the latent feature of each 3D Gaussian. This latent feature, containing anisotropic information, is then fed into a tiny feature decoding MLP $\Psi$ to determine the final specular color:

$$\Psi(\kappa, \gamma(\mathbf{d}), \langle n, -\mathbf{d} \rangle) \rightarrow c_s,$$
$$\kappa = \bigoplus_{i=1}^{N} ASG(\omega_r \mid [\mathbf{x}, \mathbf{y}, \mathbf{z}], [\lambda_i, \mu_i], \xi_i) \tag{10}$$

Table 1: **Quantitative Comparison on anisotropic synthetic dataset.**

| Dataset | Anisotropic Synthetic | | | | | |
|---|---|---|---|---|---|---|
| Method | PSNR ↑ | SSIM ↑ | LPIPS ↓ | FPS | Mem | Num.(k) |
| 3D-GS | 33.82 | 0.966 | 0.062 | 325 | 47MB | 201 |
| Scaffold-GS | 35.34 | 0.972 | 0.052 | 234 | 27MB | - |
| Ours-w/ anchor | 36.76 | 0.976 | 0.046 | 180 | 25MB | - |
| Ours-light | 37.42 | 0.979 | 0.044 | 159 | 45MB | 146 |
| Ours | 37.70 | 0.980 | 0.042 | 145 | 57MB | 183 |

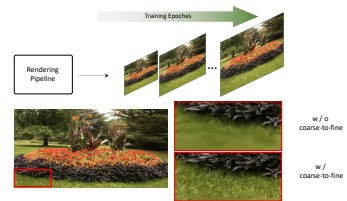

Figure 3: Using a coarse-to-fine strategy, our approach can eliminate the floaters without increasing the number of GS.

where $\kappa$ is the latent feature derived from ASG, $\bigoplus$ denotes the concatenation operation, $\gamma$ represents the positional encoding, $\mathbf{d}$ is the unit view direction pointing from the camera to each 3D Gaussian, $n$ is the normal of each 3D Gaussian, and $\omega_r$ is the unit reflect direction. This strategy significantly enhances the ability of 3D-GS to model scenes with complex optical phenomena, whereas neither pure ASG nor pure MLP can achieve anisotropic appearance modeling as effectively as our approach.

**Normal estimation.**    Following [22, 47], we use the shortest axis of each Gaussian as its normal. This approach is based on the observation that 3D Gaussians tend to flatten gradually during the optimization process, allowing the shortest axis to serve as a reasonable approximation for the normal.

The reflect direction $\omega_r$ can then be derived using the view direction and the local normal vector $n$ as:

$$\omega_r = 2(\omega_o \cdot n) \cdot n - \omega_o, \tag{11}$$

where $\omega_o = -\mathbf{d}$ is a unit view direction pointing from each 3D Gaussian in world space to the camera. We use the reflect direction $\omega_r$ to query ASG, enabling better interpolation of latent features containing anisotropic information. Experimental results show that although this unsupervised normal estimation cannot generate physically accurate normals aligned with the real world, it is sufficient to produce relatively accurate reflect direction to assist ASG in fitting high-frequency information.

### 3.3   Coarse-to-fine Training

We observed that in many real-world scenarios, 3D-GS tends to overfit the training data, leading to the emergence of numerous floaters when rendering images from novel viewpoints. One important reason is that the COLMAP point cloud is too sparse. Poor initialization makes it difficult for 3D-GS to compensate for overly sparse areas through densification during optimization, leading to floaters in the rendering images. Moreover, 3D-GS accumulates gradients from each pixel to the GS: $\frac{dL}{d\mathbf{x}} = \sum \frac{dL}{d\mathbf{p}_i} \frac{d\mathbf{p}_i}{d\mathbf{x}}$, and the densification occurs when the accumulated amount exceeds a threshold $\tau_g = 0.0002$. However, having positive and negative gradients can cause GSs that should be densified to be ignored due to the large negative gradient.

Thus, to mitigate the occurrence of floaters in real-world scenes, we propose a coarse-to-fine training mechanism. We first impose an L1 constraint on the gradients from pixels to GS: $\frac{dL}{d\mathbf{x}} = \sum \|\frac{dL}{d\mathbf{p}_i} \frac{d\mathbf{p}_i}{d\mathbf{x}}\|_1$, accumulating the numerical contribution from pixels to GS rather than gradients. This idea is similar to the concurrent works [65, 70]. Next, to avoid overfitting caused by excessive growth of 3D-GS during the early stages of optimization, we decide to train 3D-GS progressively from low to high resolution in real-world scenes:

$$r(i) = \min(\lfloor r_s + (r_e - r_s) \cdot i/\tau \rceil, r_e), \tag{12}$$

where $r(i)$ is the image resolution at the $i$-th training iteration, $r_s$ is the starting image resolution, $r_e$ is the ending image resolution (the full resolution we aim to render), and $\tau$ is the threshold iteration, empirically set to 5k.

This training method allows 3D-GS to densify correctly and prevents excessive growth of 3D-GS in the early stages. Additionally, due to the lower resolution training in the initial phase, this mechanism reduces training time by approximately 10%. In our experiments, we offer a performance version with $\tau_g = 0.0005$ and light version with $\tau_g = 0.0006$.

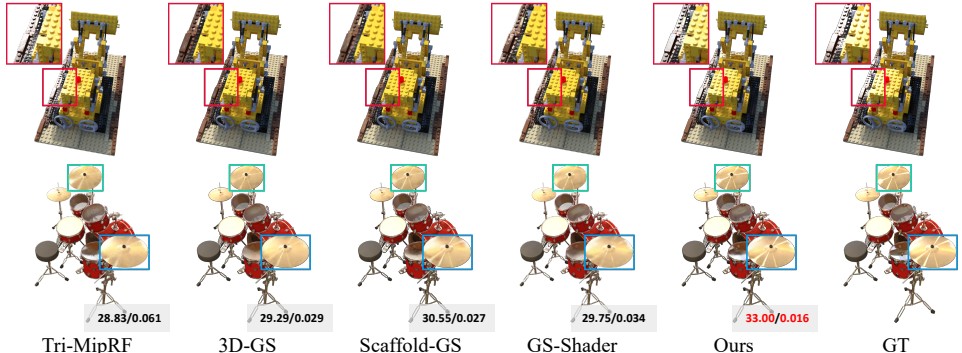

| | | | | | |
|---|---|---|---|---|---|
| 28.83/0.061 | 29.29/0.029 | 30.55/0.027 | 29.75/0.034 | 33.00/0.016 | |
| Tri-MipRF | 3D-GS | Scaffold-GS | GS-Shader | Ours | GT |

Figure 4: **Visualization on NeRF dataset.** Our method has achieved specular highlights modeling, which other 3D-GS-based methods fail to accomplish, while maintaining fast rendering speed.

| Dataset | Mip-NeRF 360 | | | | | Mip-NeRF 360 Outdoor | | | Mip-NeRF 360 Indoor | | |
|---|---|---|---|---|---|---|---|---|---|---|---|
| Method \| Metrics | PSNR ↑ | SSIM ↑ | LPIPS ↓ | FPS | Mem | PSNR ↑ | SSIM ↑ | LPIPS ↓ | PSNR ↑ | SSIM ↑ | LPIPS ↓ |
| Plenoxels | 23.08 | 0.626 | 0.463 | 6.79 | 2.1GB | 21.68 | 0.513 | 0.491 | 24.83 | 0.766 | 0.426 |
| iNGP | 25.59 | 0.699 | 0.331 | 9.43 | 48MB | 22.75 | 0.567 | 0.403 | 29.14 | 0.863 | 0.242 |
| Mip-NeRF360 | 27.69 | 0.792 | 0.237 | 0.06 | 8.6MB | 24.47 | 0.691 | 0.283 | 31.72 | 0.917 | 0.180 |
| 3D-GS | 27.79 | 0.826 | 0.202 | 115 | 748MB | 25.02 | 0.742 | 0.232 | 31.25 | 0.931 | 0.164 |
| Scaffold-GS | 27.98 | 0.824 | 0.207 | 96 | 203MB | 25.07 | 0.736 | 0.243 | 31.61 | 0.933 | 0.162 |
| Ours-w/ anchor | 28.14 | 0.824 | 0.196 | 70 | 260MB | 24.98 | 0.735 | 0.223 | 32.09 | 0.935 | 0.161 |
| Ours-light | 28.07 | 0.834 | 0.183 | 44 | 684MB | 25.09 | 0.752 | 0.203 | 31.80 | 0.936 | 0.158 |
| Ours | 28.18 | 0.835 | 0.176 | 33 | 847MB | 25.11 | 0.754 | 0.195 | 32.01 | 0.937 | 0.153 |

Table 2: **Quantitative comparison of on real-world datasets.** We report PSNR, SSIM, LPIPS (VGG) and color each cell as best , second best and third best . Our method has achieved the best rendering quality, while striking a good balance between FPS and the storage memory.

## 3.4 Adaption for Anchor-Based Gaussian Splatting

While the ASG appearance field significantly improves the ability of 3D-GS to model specular and anisotropic features, it introduces additional computational overhead due to the additional local features $\mathbf{f}$ associated with each Gaussian. Inspired by [32], we employ anchor-based Gaussian splatting to reduce storage overhead and accelerate the rendering.

Since the anisotropy modeled by ASG is continuous in space, it can be compressed into a lower-dimensional space. Thanks to the guidance of the anchor Gaussian, the anchor feature $\mathbf{f}_v$ can be used directly to compress $N$ ASGs, further reducing storage pressure. To make the ASG of neural Gaussians position-aware, we introduce the unit view direction to decompress ASG parameters. Consequently, the ASG parameters prediction in Eq. (8) is revised as follows:

$$\Theta(\mathbf{f}_v, \mathbf{d}_{cn}) \to \{\lambda, \mu, \xi\}_N, \tag{13}$$

where $\mathbf{d}_{cn}$ denotes the unit view direction from the camera to each neural Gaussian. Additionally, we set the diffuse part of the neural Gaussian $c_d = \phi(\mathbf{f}_v)$, directly predicted through an MLP $\phi$, to ensure the smoothness of the diffuse component and reduce the difficulty of convergence.

## 3.5 Losses

We optimize the learnable parameters and MLPs using the same loss function as 3D-GS [23]. The total supervision is given by:

$$\mathcal{L} = (1 - \lambda_{\text{D-SSIM}})\mathcal{L}_1 + \lambda_{\text{D-SSIM}}\mathcal{L}_{\text{D-SSIM}}, \tag{14}$$

where the $\lambda_{\text{D-SSIM}} = 0.2$ is consistently used in our experiments.

## 4 Experiments

In this section, we present both quantitative and qualitative results of our method. To evaluate its effectiveness, we compared it to several state-of-the-art methods across various datasets. We color

Table 3: **Results on NeRF synthetic dataset.**

| Dataset | NeRF Synthetic | | | | |
|---|---|---|---|---|---|
| Method \| Metrics | PSNR ↑ | SSIM ↑ | LPIPS ↓ | FPS | Mem |
| iNGP-Base | 33.18 | 0.963 | 0.045 | ~10 | 13MB |
| Mip-NeRF | 33.09 | 0.961 | 0.043 | <1 | 10MB |
| Tri-MipRF | 33.65 | 0.963 | 0.042 | ~5 | 60MB |
| 3D-GS | 33.32 | 0.969 | 0.031 | 315 | 69MB |
| GS-Shader | 33.38 | 0.968 | 0.030 | 97 | 29MB |
| Scaffold-GS | 33.68 | 0.967 | 0.034 | 240 | 19MB |
| Ours-w/ anchor | 33.96 | 0.969 | 0.032 | 172 | 19MB |
| Ours-light | 34.08 | 0.970 | 0.029 | 148 | 58MB |
| Ours | 34.19 | 0.971 | 0.028 | 121 | 72MB |

Table 4: **Results on NSVF synthetic dataset.**

| Dataset | NSVF Synthetic | | | | |
|---|---|---|---|---|---|
| Method \| Metrics | PSNR ↑ | SSIM ↑ | LPIPS ↓ | FPS | Mem |
| TensoRF | 36.52 | 0.982 | 0.026 | 1.5 | 65MB |
| Tri-MipRF | 34.58 | 0.973 | 0.030 | ~5 | 60MB |
| NeuRBF | 37.80 | 0.986 | 0.019 | ~1 | 580MB |
| 3D-GS | 37.07 | 0.987 | 0.015 | 306 | 71MB |
| GS-Shader | 33.85 | 0.981 | 0.020 | 68 | 33MB |
| Scaffold-GS | 36.43 | 0.984 | 0.017 | 218 | 17MB |
| Ours-w/ anchor | 37.71 | 0.987 | 0.015 | 152 | 16MB |
| Ours-light | 38.28 | 0.988 | 0.013 | 124 | 70MB |
| Ours | 38.40 | 0.988 | 0.012 | 108 | 89MB |

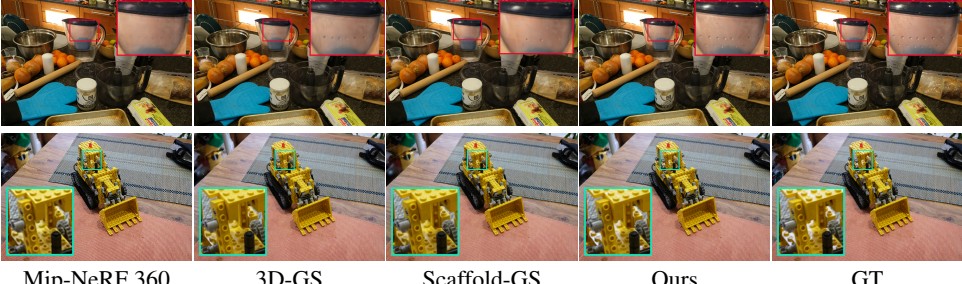

| Mip-NeRF 360 | 3D-GS | Scaffold-GS | Ours | GT |

Figure 5: **Visualization on Mip-NeRF 360 indoor scenes.** Our method achieves superior recovery of specular effects compared to SOTA methods.

each cell as best , second best and third best . Our method includes three versions, each based on different foundational methods with distinct hyperparameter settings. The performance version (Ours) is based on 3D-GS [23] with $\tau_g = 0.0005$; the light version (Ours-light), also based on 3D-GS, has $\tau_g = 0.0006$; and the mini version (Ours-w/ anchor) is based on Scaffold-GS [32], with $\tau_g = 0.0006$. Our method demonstrates superior performance in modeling complex specular and anisotropic features, as evidenced by comparisons on the NeRF, NSVF, and our "Anisotropic Synthetic" datasets. Additionally, we showcase its versatility by comparing its performance in diffuse scenarios, further proving the robustness of our approach.

## 4.1   Implementation Details

We implemented our framework using PyTorch [42] and modified the differentiable Gaussian rasterization to include depth visualization. For the ASG appearance field, the decoupling MLP $\Psi$ consists of 3 layers, each with 64 hidden units, and the positional encoding for the view direction is of order 2. Regarding coarse-to-fine training, which is applied only to real-world scenes to remove floaters, we start with a resolution $r_s$ that is **4x** downsampled. To further accelerate rendering, we prefilter and allow only those Gaussians with opacity $\sigma_n > 0$ to pass through the ASG appearance field and Gaussian rasterization pipelines. All experiments were conducted on an NVIDIA RTX 3090.

## 4.2   Results and Comparisons

**Synthetic bounded scenes.**   We used the NeRF, NSVF, and our "Anisotropic Synthetic" datasets as the experimental datasets for synthetic scenes. Our comparisons were made with the most relevant state-of-the-art methods, including 3D-GS [23], Scaffold-GS [32], GaussianShader [22], and several NeRF-based methods such as NSVF [30], TensoRF [5], NeuRBF [8], and Tri-MipRF [18].

As shown in Fig. 4 (with PSNR and LPIPS), and Tabs. 3- 4, our method achieved the highest performance with fewer Gaussians compared to vanilla 3D-GS. It also improved upon the issues that 3D-GS faced in modeling high-frequency specular highlights and complex anisotropy as shown in Tab. 1 with fewer Gaussians and better metrics. See more in the supplementary materials.

**Real-world unbounded scenes.**   To verify the versatility of our method in real-world scenarios, we used the Mip360 [2] dataset, which contains indoor scenes with specular highlights. As shown in Tab. 2, our method surpasses state-of-the-art methods on Mip-NeRF 360. Furthermore, our method effectively balances FPS, storage, and rendering quality. It enhances rendering quality without increasing storage or significantly reducing FPS. As illustrated in Fig. 5 and Fig. 7, our method has

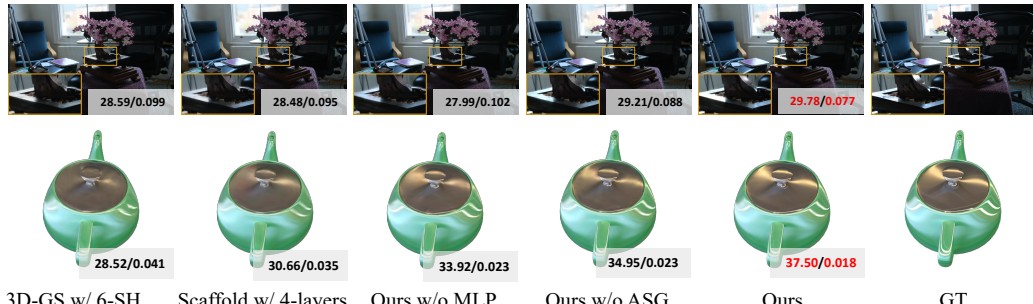

|              |                 |              |               |       |     |
| 28.59/0.099  | 28.48/0.095     | 27.99/0.102  | 29.21/0.088   | 29.78/0.077 | |
| 28.52/0.041  | 30.66/0.035     | 33.92/0.023  | 34.95/0.023   | 37.50/0.018 | |
| 3D-GS w/ 6-SH | Scaffold w/ 4-layers | Ours w/o MLP | Ours w/o ASG | Ours | GT |

Figure 6: **Ablation on ASG appearance field.** We show that directly using ASG to model color leads to the failure in modeling anisotropy and specular highlights. By decoupling the ASG features through MLP, we can realistically model complex optical phenomena.

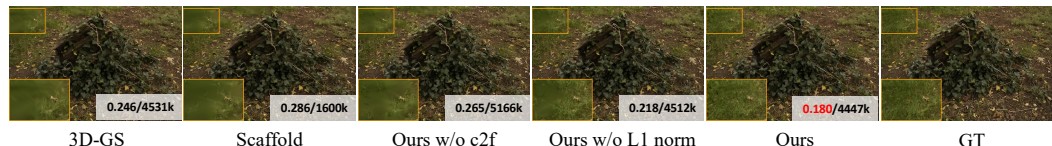

|         |          |               |                 |       |     |
| 0.246/4531k | 0.286/1600k | 0.265/5166k | 0.218/4512k | 0.180/4447k | |
| 3D-GS   | Scaffold | Ours w/o c2f  | Ours w/o L1 norm | Ours | GT |

Figure 7: **Ablation on coarse-to-fine training.** Experimental results demonstrate that our simple yet effective training mechanism can effectively remove floaters without increasing the number of 3D Gaussians, thereby alleviating the overfitting problem prevalent in 3D-GS-based methods.

also significantly improved the visual effect. It removes a large number of floaters in outdoor scenes and successfully models the high-frequency specular highlights in indoor scenes. This demonstrates that our approach is not only adept at modeling complex specular scenes but also effectively improves rendering quality in general scenarios.

### 4.3 Ablation Study

**ASG feature decoupling MLP.** We conducted an ablation study to evaluate the key components of the ASG appearance field, which include ASG features, decoupling MLP, and the separation of diffuse and specular colors. As demonstrated in Fig. 6 (with PSNR and LPIPS), directly using ASG to output color results in the inability to model specular and anisotropic components. In contrast to directly using an MLP for color modeling, as in Scaffold-GS [32], separately modeling diffuse and specular color can enhance the fitting ability for high-frequency information. ASG can encode higher-frequency anisotropic features. With the help of ASG's ability to encode high-frequency anisotropic features, the decoupling MLP can fit complex optical phenomena, leading to more accurate rendering results. We also demonstrated that higher-order SH (6-order) and more MLP layers (4-layers) do not help 3D-GS and Scaffold-GS achieve satisfactory results, highlighting the importance of ASG.

**Coarse-to-fine training.** We conducted an ablation study to assess the impact of coarse-to-fine (c2f) training. As illustrated in Fig. 7 (with LPIPS and number of Gaussian), both 3D-GS and Scaffold-GS exhibit a large number of floaters in the novel view synthesis. Coarse-to-fine training effectively reduces the number of floaters, alleviating the overfitting issue commonly encountered by 3D-GS in real-world scenarios. Applying an L1 constraint to the gradients used for 3D-GS densification further reduced the number of floaters and Gaussians. See more in the supplementary materials.

## 5 Conclusion

In this work, we introduce *Spec-Gaussian*, a novel approach to 3D Gaussian splitting that features an anisotropic view-dependent appearance. Leveraging the powerful capabilities of ASG, our method effectively overcomes the challenges encountered by vanilla 3D-GS in rendering scenes with specular highlights and anisotropy. Additionally, we innovatively implement a coarse-to-fine training mechanism to eliminate floaters in real-world scenes. Both quantitative and qualitative experiments

demonstrate that our method not only equips 3D-GS with the ability to model specular highlights and anisotropy but also enhances the overall rendering quality of 3D-GS in general scenes.

**Limitations.** Although our method enables 3D-GS to model complex specular and anisotropic features, it still faces challenges in handling reflections. Specular and anisotropic effects are primarily influenced by material properties, whereas reflections are closely related to the environment and geometry. Due to the lack of explicit geometry in 3D-GS, we cannot differentiate between reflections and materials using constraints like normals, as employed in Ref-NeRF [52] and NeRO [31]. We plan to explore solutions for modeling reflections with 3D-GS in future work.

# 6   Acknowlegements

We thank Chao Wan from Cornell University for the help during rebuttal period. This work was supported by the National Natural Science Foundation of China (Grant No. 62036010). Ziyi Yang was also supported by ByteDance MMLab.

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

# Appendix

This supplementary material provides more results that accompany the paper.

- Section A provides more ablations.
- Section B provides additional results, including more visualizations and quantitative results on complete datasets.

## A    More Ablations

In this section, we present the complete quantitative ablations on the key components of our method.

We first evaluate the role of each component of the ASG appearance field in NeRF synthetic scenes as shown in Tab. 5. The introduction of ASG improves the ability to model specular highlights and reduces the number of 3D Gaussians. The inclusion of normals did not significantly increase computational overhead, but it did enhance rendering metrics and visual quality. More importantly, we achieve better rendering quality with fewer Gaussians than vanilla 3D-GS, a characteristic that can be further explored in the future.

Next, we evaluated our full method on the Mip360 dataset in Tab. 6. It is important to note that the Mip360 dataset is divided into indoor and outdoor scenes. Indoor scenes have more specular highlights, while outdoor scenes contain a large number of floaters. The coarse-to-fine approach itself improves the quality of 3D-GS in real-world scenes, mainly by eliminating a significant amount of floaters in outdoor settings. Although the introduction of the ASG appearance field significantly increases rendering overhead, it did greatly enhance the modeling of specular highlights in indoor scenes. Under the constraints of the coarse-to-fine mechanism, our complete method combines the advantages of both, achieving the best rendering quality. To further improve rendering speed, we also implement a light version and a mini version based on Scaffold-GS. These versions offer a trade-off between rendering quality and speed and can be used as needed. The quality of the Mip360 scenes demonstrates that our method is not only capable of handling scenes with specular highlights but is also robust in real-world diffuse scenarios.

## B    More Comparisons

In this section, we present the complete quantitative results of our experiments. We report PSNR, SSIM, LPIPS (VGG), and color each cell as best , second best and third best .

### B.1    NeRF Synthetic Scenes

As shown in Tabs. 7-9, our method demonstrates the best rendering quality metrics in almost every scene. It's important to note that the experimental setup for Tri-MipRF [18] differs from other methods. It uses both the training and validation sets as training data, expanding the scale of the model's data. When its training data is limited to the training set, its metrics suffer a noticeable drop. Nevertheless, to ensure that the experimental results fully reflect the highest performance of each method, and to prevent significant drops in metrics due to differences in experimental environments, we still present the metrics from the Tri-MipRF official paper. Our method achieved more prominent metrics in scenes with notable specular reflection and anisotropy, such as Drums, Lego, and Ship. This demonstrates that our method not only improves the overall rendering quality but also has a more significant advantage in complex specular scenarios.

### B.2    NSVF Synthetic Scenes

The NSVF [30] dataset, in comparison to NeRF, features more noticeable metallic specular reflection, as presented in the Wineholder, Steamtrain, and Spaceship scenes. It is important to note that Tri-MipRF fails to converge in the Steam scene with the official code, so we did not report metrics for that scenario. As shown in Tabs. 10-12, we present the per-scene experimental results of PSNR, SSIM, and LPIPS in the supplementary material. The experimental results indicate that compared to other methods based on 3D-GS [23], our method has significant advantages in metallic highlights

Table 5: **Ablation on ASG appearance field.**

| Dataset | NeRF Synthetic | | | | |
|---|---|---|---|---|---|
| Method | PSNR ↑ | SSIM ↑ | LPIPS ↓ | FPS ↑ | Num.(k) ↓ |
| 3D-GS | 33.32 | 0.969 | 0.031 | 315 | 295 |
| w/o ASG | 34.03 | 0.969 | 0.030 | 175 | 271 |
| w/o decoup-MLP | 33.95 | 0.970 | 0.030 | 217 | 244 |
| w/o normal | 34.10 | 0.971 | 0.029 | 139 | 238 |
| Full (Light) | 34.08 | 0.970 | 0.029 | 148 | 186 |
| Full | 34.19 | 0.971 | 0.028 | 121 | 237 |

Table 6: **Ablation on full method.**

| Dataset | MipNeRF 360 | | | | |
|---|---|---|---|---|---|
| Method | PSNR ↑ | SSIM ↑ | LPIPS ↓ | FPS ↑ | Num.(M) ↓ |
| 3D-GS | 27.47 | 0.812 | 0.222 | 115 | 3.23 |
| w/o ASG field | 27.61 | 0.830 | 0.184 | 113 | 3.21 |
| w/o c2f | 28.01 | 0.823 | 0.203 | 28 | 3.56 |
| w/ anchor | 28.14 | 0.824 | 0.196 | 70 | - |
| Full (Light) | 28.07 | 0.834 | 0.183 | 44 | 2.52 |
| Full | 28.18 | 0.835 | 0.176 | 33 | 3.10 |

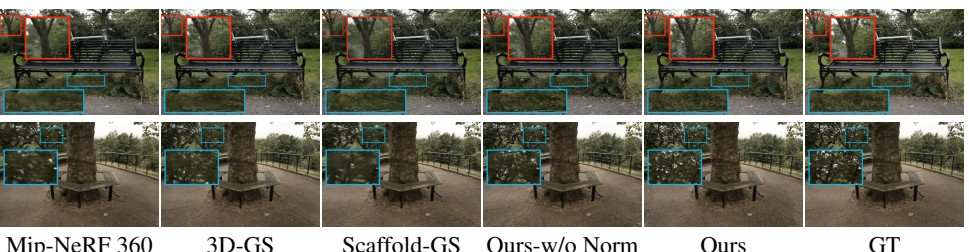

| Mip-NeRF 360 | 3D-GS | Scaffold-GS | Ours-w/o Norm | Ours | GT |

Figure 8: **Visualization on Mip-NeRF 360 outdoor scenes.** Our method achieves robust floater removal by coarse to fine training.

and complex transmission scenarios. Additionally, we compared it with the SOTA NeRF-based methods based on NeRF. Our approach enables 3D-GS to surpass the latest SOTA of NeRF, achieving high-frequency highlight modeling that 3D-GS couldn't realize but NeRF could, thereby achieving truly high-quality rendering as shown in Fig. 14.

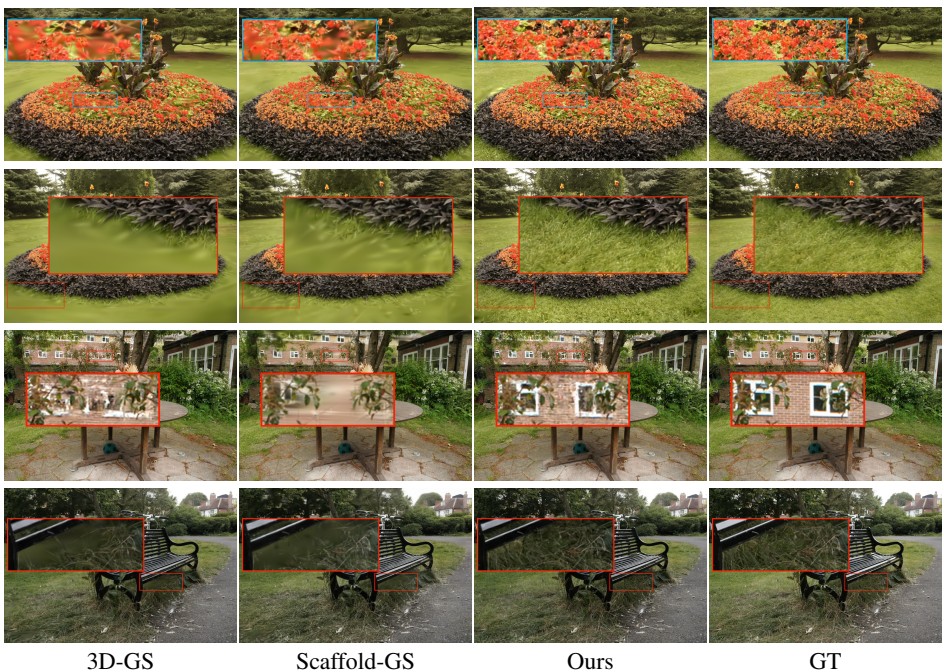

| 3D-GS | Scaffold-GS | Ours | GT |

Figure 9: **More comparisons with baselines.** Our method achieves robust floater removal by coarse-to-fine training.

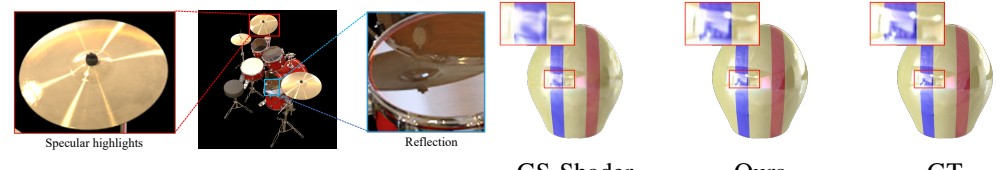

Figure 10: **Illustration of specular highlights and reflections.**

GS-Shader     Ours     GT

Figure 11: **Ccomparison on Ref-NeRF dataset.**

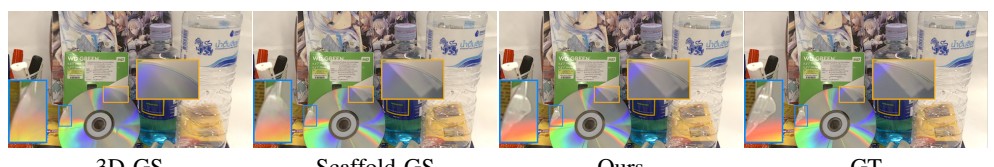

3D-GS     Scaffold-GS     Ours     GT

Figure 12: **Visualization on Nex [56] dataset.**

| | Chair | Drums | Ficus | Hotdog | Lego | Materials | Mic | Ship | Avg. |
|---|---|---|---|---|---|---|---|---|---|
| iNGP-Base | 35.00 | 26.02 | 33.51 | 37.40 | 36.39 | 29.78 | 36.22 | 31.10 | 33.18 |
| Mip-NeRF | 35.14 | 25.48 | 33.29 | 37.48 | 35.70 | 30.71 | 36.51 | 30.41 | 33.09 |
| Tri-MipRF | 36.10 | 26.59 | 34.51 | 38.54 | 36.15 | 30.73 | 37.75 | 28.78 | 33.65 |
| GS-Shader | 35.83 | 26.36 | 34.97 | 37.85 | 35.87 | 30.07 | 35.23 | 30.82 | 33.38 |
| 3D-GS | 35.36 | 26.15 | 34.87 | 37.72 | 35.78 | 30.00 | 35.36 | 30.80 | 33.32 |
| Scaffold-GS | 35.28 | 26.44 | 35.21 | 37.73 | 35.69 | 30.65 | 37.25 | 31.17 | 33.68 |
| Ours-w/ anchor | 35.57 | 26.58 | 35.71 | 38.12 | 36.62 | 30.66 | 36.81 | 31.63 | 33.96 |
| Ours-light | 35.69 | 26.77 | 36.03 | 38.25 | 36.11 | 30.84 | 36.95 | 31.97 | 34.08 |
| Ours | 35.72 | 26.92 | 36.10 | 38.25 | 36.46 | 30.98 | 37.09 | 31.97 | 34.19 |

Table 7: **Per-scene PSNR comparison on the NeRF dataset.**

| | Chair | Drums | Ficus | Hotdog | Lego | Materials | Mic | Ship | Avg. |
|---|---|---|---|---|---|---|---|---|---|
| iNGP-Base | 0.979 | 0.937 | 0.981 | 0.982 | 0.982 | 0.951 | 0.990 | 0.896 | 0.963 |
| Mip-NeRF | 0.981 | 0.932 | 0.980 | 0.982 | 0.978 | 0.959 | 0.991 | 0.882 | 0.961 |
| Tri-MipRF | 0.985 | 0.939 | 0.983 | 0.984 | 0.982 | 0.953 | 0.992 | 0.879 | 0.963 |
| GS-Shader | 0.987 | 0.949 | 0.985 | 0.985 | 0.983 | 0.960 | 0.991 | 0.905 | 0.968 |
| 3D-GS | 0.987 | 0.955 | 0.987 | 0.985 | 0.983 | 0.960 | 0.992 | 0.907 | 0.969 |
| Scaffold-GS | 0.985 | 0.950 | 0.985 | 0.983 | 0.980 | 0.960 | 0.992 | 0.898 | 0.967 |
| Ours-w/ anchor | 0.986 | 0.953 | 0.987 | 0.985 | 0.982 | 0.962 | 0.992 | 0.904 | 0.969 |
| Ours-light | 0.987 | 0.955 | 0.988 | 0.985 | 0.981 | 0.963 | 0.993 | 0.905 | 0.970 |
| Ours | 0.987 | 0.958 | 0.988 | 0.985 | 0.982 | 0.963 | 0.993 | 0.909 | 0.971 |

Table 8: **Per-scene SSIM comparison on the NeRF dataset.**

| | Chair | Drums | Ficus | Hotdog | Lego | Materials | Mic | Ship | Avg. |
|---|---|---|---|---|---|---|---|---|---|
| iNGP-Base | 0.022 | 0.071 | 0.023 | 0.027 | 0.017 | 0.060 | 0.010 | 0.132 | 0.045 |
| Mip-NeRF | 0.021 | 0.065 | 0.020 | 0.027 | 0.021 | 0.040 | 0.009 | 0.138 | 0.043 |
| Tri-MipRF | 0.016 | 0.066 | 0.020 | 0.021 | 0.016 | 0.052 | 0.008 | 0.136 | 0.042 |
| GS-Shader | 0.012 | 0.040 | 0.013 | 0.019 | 0.014 | 0.033 | 0.006 | 0.103 | 0.030 |
| 3D-GS | 0.011 | 0.037 | 0.012 | 0.020 | 0.016 | 0.037 | 0.006 | 0.106 | 0.031 |
| Scaffold-GS | 0.013 | 0.042 | 0.013 | 0.023 | 0.019 | 0.040 | 0.008 | 0.114 | 0.034 |
| Ours-w/ anchor | 0.013 | 0.038 | 0.012 | 0.022 | 0.016 | 0.037 | 0.007 | 0.112 | 0.032 |
| Ours-light | 0.012 | 0.035 | 0.011 | 0.019 | 0.017 | 0.034 | 0.006 | 0.101 | 0.029 |
| Ours | 0.011 | 0.033 | 0.011 | 0.018 | 0.014 | 0.031 | 0.006 | 0.099 | 0.028 |

Table 9: **Per-scene LPIPS (VGG) comparison on the NeRF dataset.**

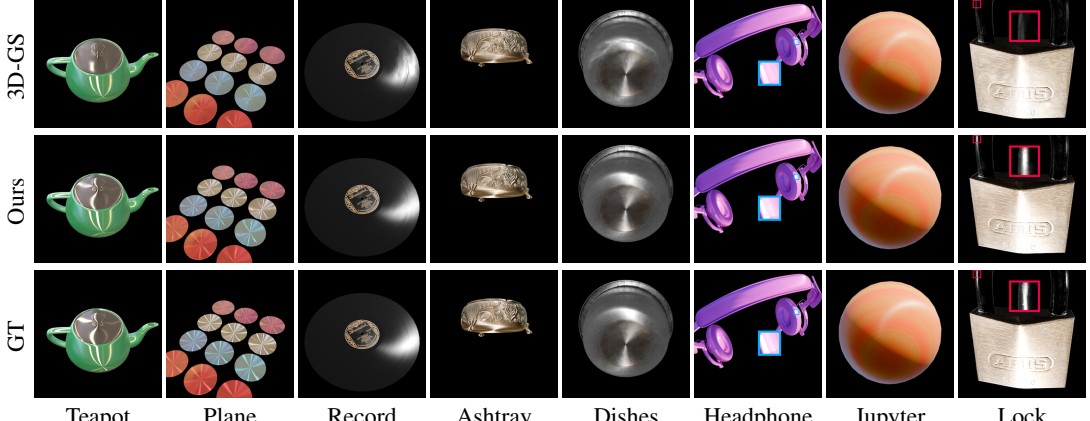

Figure 13: **Visualization on our "Anisotropic Synthetic" dataset.** We show the comparison between our method and 3D-GS across all eight scenes. Qualitative experimental results demonstrate the significant advantage of our method in modeling anisotropic scenes, thereby enhancing the rendering quality of 3D-GS.

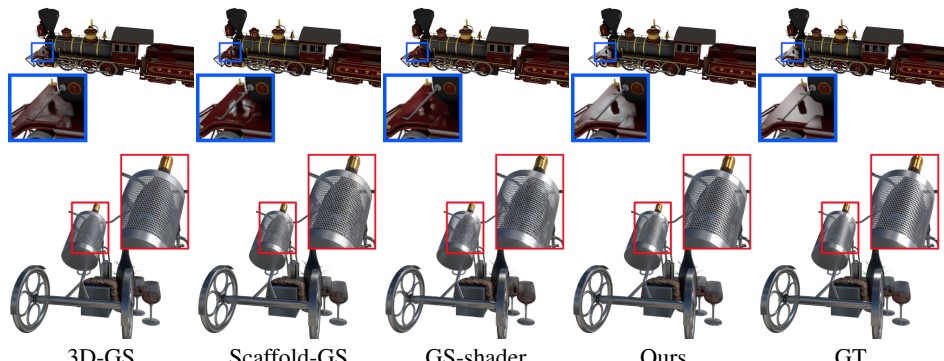

Figure 14: **Visualization on NSVF dataset.** Our method significantly improves the ability to model metallic materials compared to other GS-based methods. At the same time, our method also demonstrates the capability to model refractive parts, reflecting the powerful fitting ability of our method.

| | Bike | Life | Palace | Robot | Space | Steam | Toad | Wine | Avg. |
|---|---|---|---|---|---|---|---|---|---|
| NeRF | 31.77 | 31.08 | 31.76 | 28.69 | 34.66 | 30.84 | 29.42 | 28.23 | 30.81 |
| NSVF | 37.75 | 34.60 | 34.05 | 35.24 | 39.00 | 35.13 | 33.25 | 32.04 | 35.13 |
| TensoRF | 39.23 | 34.51 | 37.56 | 38.26 | 38.60 | 37.87 | 34.85 | 31.32 | 36.52 |
| Tri-MipRF | 36.98 | 33.98 | 36.55 | 33.49 | 37.60 | - | 33.48 | 29.97 | 34.58 |
| NeuRBF | 40.71 | 36.08 | 38.93 | 39.13 | 40.44 | 38.35 | 35.73 | 32.99 | 37.80 |
| 3D-GS | 40.76 | 33.19 | 38.89 | 39.16 | 36.80 | 37.67 | 37.33 | 32.76 | 37.07 |
| Scaffold-GS | 39.87 | 35.00 | 38.53 | 37.92 | 34.36 | 37.12 | 36.29 | 32.32 | 36.43 |
| GS-Shader | 37.38 | 27.36 | 36.55 | 37.00 | 32.61 | 35.27 | 34.50 | 30.16 | 33.85 |
| Ours-w/ anchor | 40.63 | 35.56 | 38.95 | 38.52 | 39.47 | 37.98 | 36.55 | 34.04 | 37.71 |
| Ours-light | 41.48 | 36.11 | 39.23 | 39.54 | 39.89 | 38.19 | 37.22 | 34.59 | 38.28 |
| Ours | 41.67 | 36.15 | 39.33 | 39.65 | 40.03 | 38.26 | 37.43 | 34.69 | 38.40 |

Table 10: **Per-scene PSNR comparison on the NSVF dataset.**

|        | Bike | Life | Palace | Robot | Space | Steam | Toad | Wine | Avg. |
|--------|------|------|--------|-------|-------|-------|------|------|------|
| NeRF       | 0.970 | 0.946 | 0.950 | 0.960 | 0.980 | 0.966 | 0.920 | 0.920 | 0.952 |
| NSVF       | 0.991 | 0.971 | 0.969 | 0.988 | 0.991 | 0.986 | 0.968 | 0.965 | 0.979 |
| TensoRF    | 0.993 | 0.968 | 0.979 | 0.994 | 0.989 | 0.991 | 0.978 | 0.961 | 0.982 |
| Tri-MipRF  | 0.990 | 0.962 | 0.973 | 0.985 | 0.986 | -     | 0.968 | 0.945 | 0.973 |
| NeuRBF     | 0.995 | 0.977 | 0.985 | 0.995 | 0.993 | 0.993 | 0.983 | 0.972 | 0.986 |
| 3D-GS      | 0.994 | 0.979 | 0.983 | 0.994 | 0.991 | 0.993 | 0.985 | 0.975 | 0.987 |
| Scaffold-GS| 0.993 | 0.979 | 0.981 | 0.995 | 0.985 | 0.992 | 0.982 | 0.971 | 0.984 |
| GS-Shader  | 0.992 | 0.964 | 0.979 | 0.994 | 0.985 | 0.990 | 0.980 | 0.966 | 0.981 |
| Ours-w/ anchor | 0.994 | 0.979 | 0.982 | 0.994 | 0.993 | 0.992 | 0.984 | 0.975 | 0.987 |
| Ours-light | 0.995 | 0.982 | 0.984 | 0.993 | 0.994 | 0.994 | 0.984 | 0.977 | 0.988 |
| Ours       | 0.995 | 0.982 | 0.984 | 0.995 | 0.994 | 0.994 | 0.985 | 0.978 | 0.988 |

Table 11: **Per-scene SSIM comparison on the NSVF dataset.**

|        | Bike | Life | Palace | Robot | Space | Steam | Toad | Wine | Avg. |
|--------|------|------|--------|-------|-------|-------|------|------|------|
| TensoRF    | 0.010 | 0.048 | 0.022 | 0.010 | 0.020 | 0.017 | 0.031 | 0.051 | 0.026 |
| Tri-MipRF  | 0.012 | 0.048 | 0.023 | 0.019 | 0.019 | -     | 0.036 | 0.055 | 0.030 |
| NeuRBF     | 0.006 | 0.036 | 0.016 | 0.009 | 0.011 | 0.011 | 0.025 | 0.036 | 0.019 |
| 3D-GS      | 0.005 | 0.028 | 0.017 | 0.006 | 0.009 | 0.007 | 0.018 | 0.025 | 0.015 |
| Scaffold-GS| 0.007 | 0.030 | 0.019 | 0.008 | 0.019 | 0.010 | 0.022 | 0.021 | 0.017 |
| GS-Shader  | 0.007 | 0.051 | 0.020 | 0.008 | 0.016 | 0.010 | 0.023 | 0.029 | 0.020 |
| Ours-w/ anchor | 0.005 | 0.027 | 0.018 | 0.007 | 0.007 | 0.008 | 0.021 | 0.025 | 0.015 |
| Ours-light | 0.005 | 0.024 | 0.015 | 0.006 | 0.007 | 0.007 | 0.018 | 0.022 | 0.013 |
| Ours       | 0.004 | 0.022 | 0.014 | 0.005 | 0.007 | 0.007 | 0.017 | 0.021 | 0.012 |

Table 12: **Per-scene LPIPS (VGG) comparison on the NSVF dataset.**

|        | Teapot | Plane | Record | Ashtray | Dishes | Headphone | Jupyter | Lock | Avg. |
|--------|--------|-------|--------|---------|--------|-----------|---------|------|------|
| 3D-GS      | 27.24 | 26.80 | 43.81 | 34.43 | 29.62 | 38.72 | 40.52 | 29.36 | 33.81 |
| Scaffold-GS| 30.64 | 29.14 | 47.79 | 35.66 | 32.12 | 37.19 | 40.04 | 30.13 | 35.34 |
| Ours-w/ anchor | 33.53 | 31.56 | 50.35 | 36.14 | 32.95 | 38.48 | 40.10 | 30.96 | 36.76 |
| Ours-light | 34.75 | 31.01 | 50.30 | 37.76 | 33.03 | 39.39 | 41.42 | 31.68 | 37.42 |
| Ours       | 35.24 | 30.95 | 50.90 | 38.03 | 33.04 | 40.12 | 41.47 | 31.86 | 37.70 |

Table 13: **Per-scene PSNR comparison on our "Anisotropic Synthetic" dataset.**

|        | Teapot | Plane | Record | Ashtray | Dishes | Headphone | Jupyter | Lock | Avg. |
|--------|--------|-------|--------|---------|--------|-----------|---------|------|------|
| 3D-GS      | 0.968 | 0.946 | 0.994 | 0.969 | 0.947 | 0.989 | 0.985 | 0.932 | 0.966 |
| Scaffold-GS| 0.979 | 0.965 | 0.998 | 0.973 | 0.967 | 0.986 | 0.983 | 0.924 | 0.972 |
| Ours-w/ anchor | 0.985 | 0.973 | 0.999 | 0.974 | 0.973 | 0.988 | 0.984 | 0.930 | 0.976 |
| Ours-light | 0.987 | 0.967 | 0.998 | 0.984 | 0.970 | 0.990 | 0.987 | 0.948 | 0.979 |
| Ours       | 0.988 | 0.967 | 0.999 | 0.985 | 0.970 | 0.990 | 0.987 | 0.951 | 0.980 |

Table 14: **Per-scene SSIM comparison on our "Anisotropic Synthetic" dataset.**

|        | Teapot | Plane | Record | Ashtray | Dishes | Headphone | Jupyter | Lock | Avg. |
|--------|--------|-------|--------|---------|--------|-----------|---------|------|------|
| 3D-GS      | 0.043 | 0.085 | 0.019 | 0.044 | 0.120 | 0.015 | 0.075 | 0.098 | 0.062 |
| Scaffold-GS| 0.029 | 0.057 | 0.006 | 0.038 | 0.082 | 0.021 | 0.086 | 0.099 | 0.052 |
| Ours-w/ anchor | 0.022 | 0.042 | 0.004 | 0.039 | 0.067 | 0.017 | 0.084 | 0.093 | 0.046 |
| Ours-light | 0.021 | 0.052 | 0.007 | 0.022 | 0.079 | 0.014 | 0.076 | 0.080 | 0.044 |
| Ours       | 0.021 | 0.051 | 0.005 | 0.020 | 0.077 | 0.013 | 0.071 | 0.075 | 0.042 |

Table 15: **Per-scene LPIPS (VGG) comparison on our "Anisotropic Synthetic" dataset.**

|        | bicycle | flowers | garden | stump | treehill | room | counter | kitchen | bonsai |
|--------|---------|---------|--------|-------|----------|------|---------|---------|--------|
| Plenoxels  | 21.91 | 20.10 | 23.49 | 20.66 | 22.25 | 27.59 | 23.62 | 23.42 | 24.67 |
| iNGP       | 22.17 | 20.65 | 25.07 | 23.47 | 22.37 | 29.69 | 26.69 | 29.48 | 30.69 |
| Mip-NeRF360| 24.37 | 21.73 | 26.98 | 26.40 | 22.87 | 31.63 | 29.55 | 32.23 | 33.46 |
| 3D-GS      | 25.63 | 21.94 | 27.73 | 27.02 | 22.79 | 31.80 | 29.12 | 31.61 | 32.48 |
| Scaffold-GS| 25.61 | 21.74 | 27.82 | 26.79 | 23.38 | 32.14 | 29.62 | 31.81 | 32.87 |
| Ours-w anchor | 25.44 | 21.36 | 27.97 | 26.91 | 23.23 | 32.27 | 30.12 | 32.04 | 33.91 |
| Ours-light | 25.87 | 21.81 | 28.06 | 27.23 | 22.48 | 32.11 | 29.74 | 32.09 | 33.26 |
| Ours       | 25.90 | 21.86 | 28.07 | 27.25 | 22.48 | 32.11 | 30.12 | 32.25 | 33.54 |

Table 16: **Per-scene PSNR comparison on the Mip-NeRF 360 dataset.**

| | bicycle | flowers | garden | stump | treehill | room | counter | kitchen | bonsai |
|---|---|---|---|---|---|---|---|---|---|
| Plenoxels | 0.496 | 0.431 | 0.606 | 0.523 | 0.509 | 0.842 | 0.759 | 0.648 | 0.814 |
| iNGP | 0.512 | 0.486 | 0.701 | 0.594 | 0.542 | 0.871 | 0.817 | 0.858 | 0.906 |
| Mip-NeRF360 | 0.685 | 0.583 | 0.813 | 0.744 | 0.632 | 0.913 | 0.894 | 0.920 | 0.941 |
| 3D-GS | 0.778 | 0.623 | 0.874 | 0.784 | 0.651 | 0.928 | 0.916 | 0.933 | 0.948 |
| Scaffold-GS | 0.773 | 0.609 | 0.867 | 0.774 | 0.657 | 0.931 | 0.919 | 0.931 | 0.950 |
| Ours-w anchor | 0.775 | 0.611 | 0.869 | 0.775 | 0.645 | 0.932 | 0.920 | 0.934 | 0.953 |
| Ours-light | 0.795 | 0.645 | 0.879 | 0.795 | 0.647 | 0.934 | 0.920 | 0.936 | 0.952 |
| Ours | 0.797 | 0.648 | 0.881 | 0.797 | 0.647 | 0.935 | 0.923 | 0.937 | 0.953 |

Table 17: **SSIM Comparison on the Mip-NeRF 360 dataset.**

| | bicycle | flowers | garden | stump | treehill | room | counter | kitchen | bonsai |
|---|---|---|---|---|---|---|---|---|---|
| Plenoxels | 0.506 | 0.521 | 0.386 | 0.503 | 0.540 | 0.419 | 0.441 | 0.447 | 0.398 |
| iNGP | 0.446 | 0.441 | 0.257 | 0.421 | 0.450 | 0.261 | 0.306 | 0.195 | 0.205 |
| Mip-NeRF360 | 0.301 | 0.344 | 0.170 | 0.261 | 0.339 | 0.211 | 0.204 | 0.127 | 0.176 |
| 3D-GS | 0.204 | 0.328 | 0.103 | 0.207 | 0.318 | 0.191 | 0.178 | 0.113 | 0.173 |
| Scaffold-GS | 0.224 | 0.339 | 0.112 | 0.228 | 0.315 | 0.182 | 0.177 | 0.114 | 0.174 |
| Ours-w anchor | 0.205 | 0.292 | 0.110 | 0.215 | 0.293 | 0.185 | 0.179 | 0.115 | 0.166 |
| Ours-light | 0.173 | 0.279 | 0.097 | 0.190 | 0.275 | 0.182 | 0.173 | 0.111 | 0.168 |
| Ours | 0.166 | 0.263 | 0.092 | 0.184 | 0.269 | 0.177 | 0.166 | 0.108 | 0.162 |

Table 18: **LPIPS Comparison on the Mip-NeRF 360 dataset.**

## B.3  Anisotorpic Synthetic Scenes

"Anisotropic Synthetic" is a synthetic dataset we rendered ourselves, which includes 8 scenes with significant anisotropy. We tested some existing 3D-GS-based methods on "Anisotropic Synthetic." As shown in Tabs. 13-15, our method achieved a very significant improvement in rendering metrics. Fig. 13 shows the comparison between our method and 3D-GS across all eight scenes. Qualitative experiments also demonstrate the significant visual advantages of our method, highlighting the substantial improvement our method brings to anisotropic parts, thereby enhancing the overall rendering quality.

## B.4  Mip-360 Scenes

The MipNeRF-360 scenes include five outdoor and four indoor scenarios. There are several scenes rich in specular reflections, such as bonsai, room, and kitchen. As shown in Tabs. 16-18, our method achieved significant advantages in the four indoor scenes. This reflects our method's strengths in modeling specular reflections and anisotropy. In outdoor scenes, our method also achieved rendering metrics comparable to the SOTA methods. Furthermore, with the help of the coarse-to-fine training mechanism, our method significantly reduced the number of floaters as shown in Fig. 11, resulting in a substantial improvement in visual effects.

