# OpenReview forum: "Spec-Gaussian: Anisotropic View-Dependent Appearance for 3D Gaussian Splatting"
_NeurIPS.cc/2024/Conference — NeurIPS 2024 poster_

### Official Review · Reviewer_2PwK · 2024-06-26

**Soundness:** 3
**Presentation:** 3
**Contribution:** 3
**Rating:** 6
**Confidence:** 4

**Summary:**

This paper introduces an anisotropic spherical Gaussian (ASG) appearance field into 3D Gaussian splatting for modeling the view-dependent appearance of each 3D Gaussian, which increases the ability of 3D Gaussian in representing high-frequency information. The key idea of this paper is combining ASG and SH to model the color of each 3D Gaussian. The experiments show the effectiveness of the method in modeling specular highlights and rendering quality.

**Strengths:**

1. An ASG appearance field is introduced to increase the ability of 3D Gaussian in representing high-frequency information.
2. The coarse-to-fine training scheme can effectively eliminate floaters, which is verified by the corresponding ablation studies.
3. Extensive comparison and ablation studies have been done to show the effectiveness of each component in this method.

**Weaknesses:**

1. The method of this paper includes several MLPs, however, the authors didn't report the training time of their model when compared with other methods. I think the training time should be considered for a more fair comparison. If I missed this, please correct me.
2. The structure of 3D Gaussian in this paper is mainly based on Scaffold-GS, anchor-based Gaussian splatting. Therefore, the performance of the method in this paper may be degraded when the vanilla Scaffold-GS cannot perform well, such as the scene dominated by large texture-less regions.

**Questions:**

1. It would be better if the authors could include the training time of their model when comparing it with other methods.
2. If we use RGB color as the diffuse color directly, instead of using SH to model, whether there will be a better or a worse result, since the high order SH is used in 3D-GS to represent high-frequency information.

**Limitations:**

Yes. The authors discussed the limitations of their work.

---

> ### Author Rebuttal · Authors · 2024-07-31
>
> We thank you for the positive feedback and constructive suggestions. Our response to the your  concerns are incorporated below:
>
> **Q1: Training time of our method.**
>
> This is a great question regarding the scalability of our Spec-Gaussian model. The training time of Spec-Gaussian does not significantly increase compared to the baselines. We present the comparison on NeRF-Synthetic dataset in the table below:
>
> | Method              | 3D-GS | Scaffold-GS | Ours | Ours-light | Ours-w/ anchor |
> | ------------------- | ----- | ----------- | ---- | ---------- | -------------- |
> | Training Time (min) | 8     | 9           | 15   | 14         | 11             |
>
> **Q2: Spec-Gaussian may not perform well in texture-less scenes.**
>
> It is worth noting that our approach is generic to be incorporated into 3DGS and Scaffold-GS: both the `Ours` and `Ours-light` versions are based on 3D-GS; and `Ours-w/ anchor` is based on Scaffold-GS. The anchor-based design is only used for efficiency improvements to balance the additional overhead introduced by ASG. As shown in Tab. 4, `Ours-w/ anchor` still outperforms Scaffold-GS by a large margin, demonstrating its superior modeling ability over Scaffold-GS.  Also, we didn’t observe significant degradation in texture-less areas by using anchors in NSVF dataset.
>
> **Q3: Use RGB color as the diffuse color.**
>
> Great thought. We also experimented with using RGB color as the diffuse color to reduce the computational and storage overhead introduced by spherical harmonics (SH). Experiments on the NeRF-Synthetic dataset showed that using RGB color slightly decreases the rendering metrics. Considering that Spec-Gaussian aims to explore the upper limits of 3D-GS rendering quality, we chose to filter the diffuse color with third-order SH. We will include a discussion on diffuse color modeling in the paper.
>
> | Scene    | PSNR      | SSIM      | LPIPS     | Time (3090) | FPS     | Mem    |
> | -------- | --------- | --------- | --------- | ----------- | ------- | ------ |
> | chair    | 35.75     | 0.9869    | 0.0111    | 24          | 81      | 64     |
> | drums    | 26.92     | 0.9552    | 0.034     | 17          | 113     | 46     |
> | ficus    | 35.75     | 0.987     | 0.0118    | 13          | 208     | 25     |
> | hotdog   | 38.25     | 0.9859    | 0.0184    | 14          | 156     | 31     |
> | Lego     | 36.48     | 0.9828    | 0.0156    | 14          | 137     | 38     |
> | material | 30.77     | 0.9626    | 0.0353    | 11          | 201     | 22     |
> | mic      | 36.95     | 0.9929    | 0.0059    | 13          | 153     | 26     |
> | ship     | 31.63     | 0.9037    | 0.1027    | 21          | 99      | 61     |
> | Average  | 34.07     | 0.9696    | 0.0294    | 15.87       | **144** | **39** |
> | Paper    | **34.19** | **0.971** | **0.028** | **15.42**   | 121     | 72     |

---

> ### Comment · Area_Chair_tLag · 2024-08-10
> **Please review author rebuttal!**
>
> Dear Reviewer,
>
> I wanted to gently remind you to please review the rebuttal provided by the authors. Your feedback is invaluable to the decision-making process, and if you feel that the rebuttal addresses any of your concerns, please consider updating your score accordingly.
>
> Thank you for your continued dedication to ensuring a fair and thorough review process!
>
> Best, Your AC

---

> > ### Comment · Reviewer_2PwK · 2024-08-11
> >
> > Thanks for providing the details of the training time of methods and the results using RGB color directly, which answers my questions. I also have read other reviewer's comments. I will keep my initial rating.

---

### Official Review · Reviewer_8TqS · 2024-07-09

**Soundness:** 3
**Presentation:** 3
**Contribution:** 3
**Rating:** 5
**Confidence:** 4

**Summary:**

Spherical harmonics-based 3D Gaussian splatting (3DGS) struggles with specular and anisotropic components. To address this problem, the paper proposes adopting anisotropic spherical Gaussians (ASG). However, directly adopting ASG does not demonstrate superior performance in representing specular and anisotropic parts. Therefore, the paper proposes separating diffuse and specular components from color representations and using a feature decoupling MLP to generate colors from ASG features. Through experiments, the paper demonstrates improved ability to represent highly specular parts.

**Strengths:**

**Novelty**

The idea of adopting ASG for color representation in the Gaussian Splatting framework is novel.



**Performance**

The paper demonstrates significant performance improvements across different datasets.

**Weaknesses:**

**Related Works**

The related works section could better highlight the differences and advantages of this paper compared to other studies. The paper does not mention other works on specular scenes and objects. For instance, GaussianShader is another 3D Gaussian splatting-based method for specular scenes and objects, yet it is not mentioned in the related works section, even though it is referenced elsewhere in the paper. The related works section should compare this method with prior works, such as SpecNeRF (CVPR 2024), and highlight their limitations and how the proposed method overcomes them, or emphasize the novelty of this paper.





**Mathematical Notations**

The mathematical notations could be improved.
The inner product is represented both by $\cdot$ (Line 145, Eq. 11) and $\langle \rangle$ (Eq. 10).
Additionally, $\cdot$ denotes element-wise multiplication (Eqs. 5 and 6), inner product (Line 145, Eq. 11—inside the parenthesis), scalar-vector multiplication (Eq. 11—outside the parenthesis), and scalar-scalar multiplication (Eqs. 4 and 12).
At least element-wise multiplication and inner product should use different notations.
And using the same notation for different operations within a single equation (Eq. 11) should also be avoided.





**Coarse-to-fine Training**

Coarse-to-fine training is not a novel approach within the 3DGS framework.
For example, “EAGLES: Efficient Accelerated 3D Gaussians with Lightweight EncodingS” already proposed a similar coarse-to-fine training approach.

**Questions:**

Line 144: Is $\xi$ really $\mathcal{R}^2$ or is this a typo of $\mathcal{R}$?



Lines 185: Could you specifically state what is decoupled through $\psi$.

Line 191: There are no equations for pure ASG nor pure MLP. Could you state the equations or clarify how they work? For example, for pure MLP, is $\kappa$ removed from Eq. 10 and only $\Psi(\gamma(d), \langle n, -d\rangle)$ used?



Lines 232-234: This seems contradictory to lines 178-179. Could you clarify which one was actually used during the experiments.



Lines 242-244 and Tabs. 3 and 4: Based on the explanation, Ours-light refers to 3DGS + ASG, while Ours-w/ anchor refers to Scaffold-GS + ASG. However, it is unclear what the performance version (Ours) refers to. As stated in lines 220 and 221, is the performance version the same as the light version (3DGS + ASG) with a lower threshold $\tau_g$? If so, clarifying this in the experimental section would help readers understand what the performance version is.



Fig 6: Lines 181-183 and 277 state that directly using ASG leads to an inability to represent specular and anisotropic components. If “Ours w/o MLP” is the one, clarifying this in the caption could help with understanding. In addition, the meaning of “MLP” in “Ours w/o MLP” is ambiguous. It is unclear whether MLP in Fig. 6 refers only to $\Psi$ (Eq. 10) or both $\Psi$ and $\phi$ (line 233).




**Color separation**

As shown in Eq. 9, the proposed method separates diffuse and specular components.
What happens without color separation? Additionally, what happens if the diffuse part is also represented through ASG? Based on the ASG paper, replacing SH with ASG can improve performance. Therefore, could you provide a reason why the diffuse part still uses SH?

**Limitations:**

The paper stated its limitation.

---

> ### Author Rebuttal · Authors · 2024-07-31
>
> We thank the reviewer for the positive and detailed review as well as the suggestions for improvement. We will revise the mathematical notations in the paper based on these insightful suggestions. Our response to the reviewer’s comments is below:
>
> **Q1: Color Separation.**
>
> Great question. It's worth noting that in Fig. 6, we've already explored the model without separation. This is because Scaffold-GS inherently uses an MLP to represent color without distinguishing between diffuse and specular components. We conducted further experiments on the teapot scene in Fig. 6, as shown in the table below:
>
> | Method | 3D-GS (SH color) | 3D-GS (MLP color) | Scaffold-GS (MLP color) | Ours (SH diffuse + ASG specular) | Ours (ASG diffuse + ASG specular) | Ours-w/ anchor (SH diffuse + ASG specular) |
> | ------ | ---------------- | ----------------- | ----------------------- | -------------------------------- | --------------------------------- | ------------------------------------------ |
> | PSNR   | 27.04            | 11.32 (Failed)    | 32.46                   | 37.50                            | 33.93                             | 35.69                                      |
>
> It can be observed that color separation significantly improves the rendering quality of scenes with specular highlights, while using ASG to model diffuse color results in a decline in rendering quality. This experimental result reveals that clean separation of diffuse and specular components can aid in the learning of each part, thereby enhancing the overall color quality. Compared to ASG, SH serves as a better **low-frequency filter**. The inherent limitations in SH's fitting capability actually contribute to a cleaner diffuse component, allowing ASG to focus more effectively on learning the specular part.
>
> **Q2: Related work.**
>
> We will add and discuss them in the related work. GaussianShader cannot effectively address scenes with specular highlights; it primarily attempts to enhance the capability of 3D-GS in modeling reflective scenes.
>
> We are sorry for missing Spec-NeRF. Although we both aim to address specular highlights, the approaches to modeling specular highlights differ from each other: Spec-NeRF uses Gaussian directional encoding, while we employ anisotropic spherical Gaussian (ASG).
>
> **Q3: Coarse-to-fine training.**
>
> Thank you very much for the reminder. We have taken note of the outstanding work in EAGLES, and we will cite this paper in Spec-Gaussian. It is important to note that, unlike EAGLES, our coarse-to-fine approach includes two components: 1) L1-normed gradients for GS densification, and 2) progressively training from low to high resolution. This design aims to prevent GS from becoming overly densified in the early stages of training (due to the L1 norm), which significantly reduces the number of GS.
>
> **Q4: Explanation of other questions.**
>
> > Line 144: Is $\xi$ really $\mathbb R^2$ or is this a typo of $\mathbb R$?
>
> - $\xi$ is $\mathbb R^2$, which means it is a 2-dimensional real vector.
>
> > Lines 185: Could you specifically state what is decoupled through $\Psi$.
>
> - Thank you for raising this question. We now believe that `decode` better conveys the meaning of $\Psi$ compared to `decouple`. Specifically, it involves decoding the ASG-encoded features to obtain the specular color.
>
> > Line 191: There are no equations for pure ASG nor pure MLP. Could you state the equations or clarify how they work?
>
> - The ablation of pure ASG and pure MLP aims to demonstrate that both ASG and decode MLP are crucial. In the case of pure ASG, we directly employ ASG to obtain the color through $c_s = \bigoplus_{i=1}^{N} ASG(\omega_r \: | \: [\mathbf{x}, \mathbf{y}, \mathbf{z}],  [\lambda_i, \mu_i], \xi_i), \text{where}\  \xi_i \in \mathbb R^3$. While for the pure MLP, we need to input features that have not been encoded by ASG to ensure a fair comparison. Therefore, the formula is shown below:  $\Psi (\kappa, \gamma(\mathbf{d}), \langle n, -\mathbf{d} \rangle) \rightarrow c_s, \kappa = \bigoplus_{i=1}^{N} [\lambda_i, \mu_i, \xi_i].$
>
> > Lines 232-234: This seems contradictory to lines 178-179. Could you clarify which one was actually used during the experiments.
>
> - As previously mentioned, our method has three versions. `Ours` and `Ours-light` are based on 3D-GS, while `Ours-w/ anchor` is based on Scaffold-GS. For the versions based on 3D-GS, we used the approach described in Lines 178-179, and for the version based on Scaffold-GS, we adopted the color model described in Lines 232-234.
>
> > Lines 242-244 and Tabs. 3 and 4: Based on the explanation, Ours-light refers to 3DGS + ASG, while Ours-w/ anchor refers to Scaffold-GS + ASG. However, it is unclear what the performance version (Ours) refers to.
>
> - Thank you for pointing out this issue. Clarifying this in the experimental section is very important. We will include the previous explanation in the paper.
>
> > Fig 6: Lines 181-183 and 277 state that directly using ASG leads to an inability to represent specular and anisotropic components. The meaning of “MLP” in “Ours w/o MLP” is ambiguous.
>
> - As mentioned in Lines 181-183, directly using ASG can result in failure to model specular color. Therefore, in the ablation study of Fig. 6, we included `Ours-w/o MLP` to support this statement. Here, MLP refers only to the decoding MLP $\Psi$ in Eq. (10). This is an excellent suggestion, and we will include a more detailed explanation of this in the paper.
>
> Finally, we would like to thank the reviewer once again. Many of these points were things we had not noticed before, and these suggestions will significantly improve the readability of the paper.

---

> > ### Comment · Reviewer_8TqS · 2024-08-09
> >
> > I appreciate the authors for their detailed rebuttal. It has addressed concerns to some extent. However, I agree with reviewer anMP that the writing could be improved. Specifically, the rebuttal did not address the following point: "In section 3, it is unclear which components belong to the base method and which are part of the method variants." Clarifying these details could strengthen the paper.

---

> ### Author Response · Authors · 2024-08-09
> **Official Comment by Authors**
>
> Thank you for the prompt response and your suggestions.
>
> As mentioned in the global response, our method has three variants: the 3D-GS-based `Ours` and `Ours-light`, and the Scaffold-GS-based `Ours-w/ anchor`.
>
> In Section 3,
> - Sec 3.1 is the Preliminaries, where 3D Gaussian splatting and Anchor-based Gaussian splatting are introduced as explanations of 3D-GS and Scaffold-GS, respectively. The Anisotropic Spherical Gaussian is introduced to explain the ASG formula, which will be applied to each of our variants to encode specular features.
> - Sec 3.2 is the modeling of view-dependent appearance for `Ours` and `Ours-light`.
> - Sec 3.4 is the modeling of view-dependent appearance for `Ours-w/ anchor`, which is slightly different from Sec 3.2.
> - Sec 3.3 is a general part used for all three variants of our method, aimed at removing floaters in real-world scenes.
>
> We hope our explanation can resolve your confusion about Section 3. And we fully agree that clarifying these details will strengthen the readability of the paper. We will include more detailed explanations for Section 3 in the paper.

---

> > ### Comment · Reviewer_8TqS · 2024-08-11
> >
> > I appreciate you for further addressing my concern. Thank you.

---

### Official Review · Reviewer_Ye8M · 2024-07-11

**Soundness:** 3
**Presentation:** 3
**Contribution:** 3
**Rating:** 6
**Confidence:** 5

**Summary:**

This paper proposes using Anisotropic Spherical Gaussians (ASGs) as view encoding to enhance the modeling of specular reflections in 3D Gaussian splatting. In addition to Spherical Harmonics (SH) encoded colors, the method additionally queries reflection direction with multiple ASGs to generate a view encoding. Since ASGs can potentially model higher frequency signals, the proposed method improves reconstruction quality in scenarios with strong view-dependence. The paper also introduces minor contributions, such as coarse-to-fine training and ASG compression, to further enhance quality and efficiency. The results demonstrate that the method surpasses all baselines.

**Strengths:**

The idea presented is neat and simple, which is good. Although it is not too surprising that introducing higher frequency view encoding could enhance appearance modeling, it is noteworthy that no one else has systematically explored this idea. This could inspire the community. Thus, there are contributions, though not significant. I also appreciate the effort to improve rendering and memory efficiency after incorporating the ASG model.

**Weaknesses:**

• Contribution: I would not describe this work as "the first to address specular highlights modeling in GS," as some earlier work has also attempted this. It would be better to tone down this claim.

• Sum of L1 Norm of Gradients: One improvement in the paper is achieved by adding L1 norm to the gradient accumulation for GS densification, as illustrated in lines 202-214. I don't really follow the intuition behind this design. The original design for densification involves duplicating Gaussians and moving along the gradient direction to reduce image loss from all views. So when different views suggest moving the Gaussian in different directions, these directions cancel out and we don't do densification. This makes sense because it means that the particular gaussian is centered at an optimal location that won't sacrifice the quality of any view. However, I cannot find any physical meaning for using the sum of L1 norms. Although experiments show better performance, it would be helpful to provide an explanation, possibly with a toy example. Could this performance improvement be because the sum of L1 norms tends to be larger and thus more likely to exceed the threshold? Would reducing the threshold have a similar effect?

• Lack ablation on the Number of ASGs: An important hyperparameter is the number of ASGs used. Theoretically, this is crucial for balancing efficiency and quality.

• Discussion on Shape-Radiance Ambiguity: An important aspect of reflection modeling in radiance fields is dealing with shape-radiance ambiguity. I assume the coarse-to-fine training helps with this ambiguity, but it is not discussed in the methods or experiments.

• Missing Comparison with Inverse Rendering GS: Although inverse rendering has a slightly different task than view-dependent appearance modeling, it is a valid approach to solving this problem. Therefore, it is necessary to compare this method with at least one GS inverse rendering method to demonstrate the benefits of using ASG over IR.

• Rendering Time Breakdown: It would be helpful to show a breakdown of rendering time for each sub-step to identify the bottleneck preventing the model from achieving a similar FPS as the original GS.

• Additional Visualizations: I would like to see visualizations of the learned ASGs and normal maps.

• Performance on Ref-NeRF's In-the-Wild Data: I am curious to see how this method performs on Ref-NeRF's in-the-wild data, which contains more challenging mirror-like reflections.

• Missing References: Some important citations are missing. Here are a few:
		○ Reflection Modeling in 3DGS:
			§ "3D Gaussian Splatting with Deferred Reflection"
		○ Reflection Modeling in Point Cloud:
			§ "Neural Point Catacaustics for Novel-View Synthesis of Reflections"
		○ Inverse Rendering in NeRF:
			§ "PhySG: Inverse Rendering With Spherical Gaussians for Physics-Based Material Editing"
		○ Reflection Modeling in NeRF:
			§ "NeRF-Casting: Improved View-Dependent Appearance with Consistent Reflections"
			§ "SpecNeRF: Gaussian Directional Encoding for Specular Reflections"

**Questions:**

Please see the weakness.

**Limitations:**

The modeling of complex reflections (e.g., self-reflections, mirror-like reflections) could be further discussed.
Also it would be helpful to demonstrate some failure cases.

---

> ### Author Rebuttal · Authors · 2024-08-01
>
> We thank the reviewer for the positive review as well as the insightful suggestions for improvement. Our response to the reviewer’s comments is below:
>
> **Q1: Ablation on the Number of ASGs.**
>
> That's an excellent question. During the implementation of our code, we explored the number of ASGs. In the table below, we present the ablation results on the `teapot` scene. ASG=32 achieves the highest overall rendering metrics without causing a significant increase in training time or a decrease in FPS. Further increasing the numbers will not result in increase of rendering qualities but decreasing the rendering speed.
>
> |    ASG Num     |   PSNR    |    SSIM    |   LPIPS    | Training Time (min) | FPS     | Mem (MB) |
> | :------------: | :-------: | :--------: | :--------: | :-----------------: | ------- | -------- |
> |       8        |   31.93   |   0.9814   |   0.0281   |       **11**        | **185** | **37**   |
> |       16       |   34.43   |   0.9861   |   0.0224   |         12          | 147     | 39       |
> | **32 (paper)** | **35.24** | **0.9876** | **0.0206** |         13          | 153     | 38       |
> |       64       |   34.81   |   0.9872   |   0.0209   |         15          | 111     | 41       |
>
> **Q2: Comparison with inverse rendering GS.**
>
> Although theoretically, it is reasonable to incorporate inverse rendering to improve rendering quality, this suffers difficulties and may cause burdens in practice. This is because decoupling and learning the required information for physically based rendering (PBR) from multiview images is a highly ill-posed problem. The errors will, in turn, have negative impacts on rendering. The table below compares our method with Relightable-GS on the NeRF-Synthetic dataset, and Spec-Gaussian outperforms Relightable-GS by a large margin.
>
> |         Scene |   PSNR    |   SSIM    |   LPIPS   |
> | ------------: | :-------: | :-------: | :-------: |
> |         chair |   33.19   |  0.9798   |  0.0177   |
> |         drums |   25.41   |  0.9484   |  0.0444   |
> |         ficus |   32.64   |   0.979   |  0.0195   |
> |        hotdog |   35.56   |  0.9802   |  0.0291   |
> |          lego |   34.37   |  0.9776   |  0.0216   |
> |     materials |   28.50   |  0.9488   |  0.0474   |
> |           mic |   33.94   |  0.9863   |  0.0134   |
> |          ship |   29.63   |  0.8891   |  0.1226   |
> |       Average |   31.66   |  0.9612   |  0.0394   |
> | Spec-Gaussian | **34.19** | **0.971** | **0.028** |
>
> **Q3: Missing comparison and citation about works on reflection.**
>
> Thanks for mentioning this. Although reflection and specular highlights (ours) focus on two different properties of objects and environments for causing view-dependent appearances as clarified previously, we will cite these works and discuss their differences in the related work. Although our work is not directly related to addressing reflection, we have still provided some experimental results for comparison. The qualitative experimental results can be seen in the submitted rebuttal PDF. The table below shows the comparison on real-world scenes from ref-nerf.
>
> |          | PSNR      | SSIM       | LPIPS      |
> | -------- | --------- | ---------- | ---------- |
> | garden   | 23.11     | 0.6174     | 0.1677     |
> | sedan    | 26.42     | 0.7317     | 0.1442     |
> | toy      | 24.93     | 0.6539     | 0.1454     |
> | Average  | **24.82** | **0.6677** | 0.1524     |
> | Ref-NeRF | 24.45     | 0.6650     | **0.1478** |
>
> We also provided a comparison between Spec-Gaussian and GaussianShader on the Ref-NeRF Synthetic scenes (Shiny-Blender):
>
> | Scene         | PSNR      | SSIM       | LPIPS      | FPS     |
> | ------------- | --------- | ---------- | ---------- | ------- |
> | Spec-Gaussian | **31.00** | 0.9500     | **0.0752** | **145** |
> | GS-Shader     | 30.73     | **0.9540** | 0.0798     | 87      |
>
> We would like to emphasize once again that specular highlights and reflections are two distinct shading effects, each requiring different technical approaches to address. Improvement in one shading effect does not necessarily translate to improvement in the other. Generally speaking, enhancing the modeling of specular highlights can also improve the modeling capability for general scenes. However, improving reflections mainly enhances the rendering quality of the reflective parts, and it may lead to negative optimization for the non-reflective parts, like NeRO and GS-DR.
>
> **Q4: Missing references.**
>
> Thank you for pointing out these awesome works. We will add these citations to our paper.
>
> **Q5: About the insights of the L1 norm of gradients.**
>
> During the optimization process, 3D-GS accumulates the gradient of each pixel ( $\frac{d L}{d \mathbf{x}}=\sum \frac{d L}{d \mathbf p_i} \frac{d \mathbf p_i}{d \mathbf{x}}$) for every GS. When the accumulated value exceeds a threshold $\tau_g$, the GS will densify. It is important to note that this value is not the gradient of each GS position but rather the accumulated gradient sum used for densification. Our insight is that gradients can be both positive and negative, and summing them for accumulation is not reasonable because large negative gradients can decrease the accumulated value, preventing GS that should densify from doing so. While negative gradients are meaningful for position optimization, they are clearly not reasonable for accumulation to determine whether densification should occur, as large negative gradients indicate that the GS requires more refined optimization. Therefore, we decided to apply the `L1 norm` **only** to the gradients used for accumulation to determine whether densification should occur ( $\frac{d L}{d \mathbf{x}}=\sum \Vert \frac{d L}{d \mathbf p_i} \frac{d \mathbf p_i}{d \mathbf{x}} \Vert_1$).
>
> **Q6: Time breakdown.**
>
> We have provided the impact of different components on FPS in the paper. It can be found in Tabs. 5-6.

---

> > ### Comment · Reviewer_Ye8M · 2024-08-11
> >
> > I appreciate the effort authors put into the rebuttal. Overall I'm satisfied with the rebuttal and it addresses most of my concerns. However, here are a few further comments:
> >
> > 1. It would be helpful to explain why further increasing the number of ASGs does not lead to increasing of rendering qualities, which is somehow counter-intuitive
> >
> > 2. Regarding the L1 norm of gradients, I believe it's totally reasonable to accumulate negative gradients both for optimization and densification, as the goal of the densification is to aid in optimization. My understanding is that accumulating L1-normed gradient tends to results in larger accumulation and could trigger densification more easily. So one experiment that may worth trying is to lower the threshold for densification, which could have a similar effect. One scenario where using L1-norm may be beneficial is in cases of view inconsistency, where a 3D Gaussian may have gradient in different directions. In this situation, using L1-norm will tends to densify and help to explain the view inconsistency with more gaussians. But in any case, simply stating that the original method is unreasonable without further explanation seems vague and potentially confusing.

---

> > > ### Author Response · Authors · 2024-08-11
> > >
> > > Thanks so much for the constructive feedback. We hope that our response below will address your concerns.
> > >
> > > **Q1: Why increasing the number of ASGs does not lead to increasing of rendering qualities.**
> > >
> > > As shown in Eq. 10, the number of ASGs affects the input dimension of the decoding MLP. While increasing the number of ASGs theoretically enhances the encoding capability of specular features, it may also lead to overfitting in the decoding MLP. This can reduce the model's generalization ability, resulting in a decline in rendering metrics on the test set.
> > >
> > > **Q2: More explanations about L1-normed gradients.**
> > >
> > > Thanks to the reviewer for the in-depth analysis. We would like to explain the rationale behind using the L1-norm from two perspectives.
> > > - From a theoretical standpoint, L1-norm gradients can alter the distribution of densification. This allows regions that need densification to be correctly densified without producing floaters, while regions that do not need densification can avoid excessive growth, thereby reducing memory overhead. This is something that simply lowering the densification threshold cannot achieve.
> > >
> > > - From an experimental perspective, we conducted experiments where we lowered the threshold, with $\tau_g=0.0002$ being the densification threshold for vanilla 3D-GS.
> > > | Method                               | PSNR  | SSIM  | LPIPS | Mem  | FPS  |
> > > | ------------------------------------ | ----- | ----- | ----- | ---- | ---- |
> > > | Ours (w/ L1 & $\tau_g=0.0005$)       | 28.18 | 0.835 | 0.176 | 848  | 33   |
> > > | Ours-light (w/ L1 & $\tau_g=0.0006$) | 28.07 | 0.834 | 0.183 | 684  | 44   |
> > > | Ours (w/o L1 & $\tau_g=0.0001$)      | 28.12 | 0.831 | 0.187 | 1619 | 18   |
> > > | Ours (w/o L1 & $\tau_g=0.0002$)      | 28.05 | 0.828 | 0.194 | 1044 | 26   |
> > >
> > > The experimental results show that our method can improve rendering metrics without increasing memory usage, demonstrating that the L1 norm is more effective than simply lowering the threshold. Beyond the improvement in metrics, the more important observation is that lowering the threshold still results in visual floaters, whereas the L1-norm can effectively remove them.

---

> > > > ### Comment · Reviewer_Ye8M · 2024-08-11
> > > >
> > > > Thanks for the additional explanation and experiments. I have no further comments.

---

> ### Comment · Area_Chair_tLag · 2024-08-10
> **Please review author rebuttal!**
>
> Dear Reviewer,
>
> I wanted to gently remind you to please review the rebuttal provided by the authors. Your feedback is invaluable to the decision-making process, and if you feel that the rebuttal addresses any of your concerns, please consider updating your score accordingly.
>
> Thank you for your continued dedication to ensuring a fair and thorough review process!
>
> Best, Your AC

---

### Official Review · Reviewer_anMP · 2024-07-11

**Soundness:** 3
**Presentation:** 1
**Contribution:** 2
**Rating:** 6
**Confidence:** 4

**Summary:**

This paper presents an approach for reconstruction and view synthesis of scenes that exhibit strong specular/view dependent appearance. In particular, the authors extend the framework of Gaussian Splatting [Kerbl et al. 2023] and Scaffold-GS [Lu et al. 2023], replacing spherical harmonics for parameterizing view-dependent appearance with anisotropic spherical Gaussians [Xu et al. 2013], as well as leveraging a coarse-to-fine training strategy to generally improve performance. The authors carry out qualitative/quantitative evaluation on a variety of datasets, and report improved quantitative performance over nearly every other baseline.

**Strengths:**

The authors show good quantitative performance on a large number of datasets -- the quantitative evaluation in particular is quite comprehensive -- and some of the qualitative results presented in the video/paper are compelling. The design of the method seems sensible, and various components are ablated to show their importance.

The largest contribution of the work (and an important one, if accurate) is assembling a system for reconstruction and view synthesis that performs well for challenging view dependent scenes.

**Weaknesses:**

* I felt that the writing quality could be improved. In section 3, I couldn't tell which of the described components were part of the base method, and which were part of method variants (e.g. ours-light, ours w/ anchor). Method training and architecture details were somewhat sparse (only a few details were provided in section 4.1). I was also confused about how the word anisotropy is being used throughout the paper. Typically, it's used to mean view dependence that is not isotropic (e.g. rotationally invariant) -- but it does necessarily imply *high frequency* view dependence. Spherical harmonics are, in fact, anisotropic spherical functions.
* I'm not quite sure how to assess the novelty of this paper, and feel that some claims (e.g. this is "the first work to address the specular highlights modeling in 3D-GS") are not quite fair (other works, such as GaussianShader [Jiang et al. 2024] at CVPR 2024 attempt to improve modeling of view dependent appearance in Gaussian splatting).
* For many of the real scenes, it's hard to judge the qualitative improvement of this method over baselines (e.g. Figure 8). The video comparisons are nice, but are not provided for all baselines -- I feel that a webpage would've been more effective for showcasing comparisons/improvements.
* As far as I can tell, this work does not make many changes on top of existing methods -- the two main changes being using ASGs to parameterize view-dependent appearance, and implementing a coarse-to-fine training strategy. Perhaps I'm missing something, but I'm not sure why these changes should lead to such a large improvement in view-dependent appearance modeling and quantitative performance. For example, are ASGs really responsible for modeling the reflection in the CD in Figure 9? Or is the ability to model this reflection due to Scaffold-GS's view-dependent decoding of Gaussian parameters? When the removal of floaters is shown in the video for the Bonsai scene, is this due to the coarse-to-fine training strategy, or something else (hard to say, because only the baseline 3DGS is shown in the comparison)? In general I don't feel that the relationship between quantitative/qualitative gains and method design are fully justified, but I acknowledge that I could be in the minority here.

**Questions:**

See above.

**Limitations:**

Limitations are discussed.

---

> ### Author Rebuttal · Authors · 2024-08-01
>
> We are glad and appreciate that you recognizes that the results of Spec-Gaussian are comprehensive and compelling. Our response to your valuable comments is below:
>
> **Q1: What makes Spec-Gaussian work: Evaluation of the different components.**
>
> The key components that make Spec-Gaussian work includes: the ASG appearance field and the coarse-to-fine training mechanism. These components are extensively studied both quantitatively (Tables 5-6) in supplementary file and qualitatively in Figures 6-8.
>
> First, the ASG appearance field works by using ASG and decoding MLP to augment Gaussians capabilities. (a) With the ASG appearance field, the performance improves by ~4dB in our anisotropic synthetic dataset. Our extensive ablation study of the ASG component also demonstrates that the improvements are not from the using of MLP like in Scaffold-GS (see Figure 6 and Tables 1-4), but more fundamentally from the introduction of ASG to effectively capture the high-frequency specular colors. (b) Moreover, our empirical experiments also show that using SH or MLP to fit the entire color spectrum independently is not ideal, as color contains both high and low-frequency signals, making it difficult to fit accurately. By filtering diffuse with SH and modeling the remaining specular components with ASG, each part can operate within its fitting capability, thereby improving the overall rendering quality.
>
> Second, the coarse-to-fine strategy is effective in resolving the floaters in real-world scenes. The reason might be that by fitting the model on coarse images, the model is encouraged to capture the low-frequency geometry instead of the high-frequency noisy details, which reduces the chance for overfitting to noisy details that are harmful for generalization causing floaters. We also introduced L1-normed gradients in the accumulation process used for densification, making the GS densification more reasonable.
>
> **Q2: Real-world comparison with Scaffold-GS.**
>
> In this rebuttal, we have submitted more comparisons incorporating scaffold-GS for comparisons and a zoomed-in version of Fig. 8 in the PDF.  We will incorporate more comparisons in the final version.
>
> **Q3: About the term `Anisotropic`.**
>
> In this paper, the term `anisotropic` often appears alongside `specular`. Take Figure 1's CD as an example—the anisotropic part refers to the CD's surface, which shows different specular colors when viewed from different angles (resulting in the rainbow effect). While SH is indeed an anisotropic spherical function, using low-order SH (e.g., first 3 orders in 3D-GS) struggles to model complex shading effects. In Figure 6, even the first 6 orders of SH falls far short of properly modeling specular scenes (perhaps using more than 100 orders would make a difference, but the computational cost would be far greater than that of ASG).

---

> > ### Comment · Reviewer_anMP · 2024-08-12
> >
> > I thank the authors for the rebuttal. It addressed many of my questions, e.g.: (1) what kinds of effects the ASGs are supposed to model, and (2) which qualitative improvements each contribution (ASGs, coarse-to-fine training) is responsible for.
> >
> > I think, perhaps, I judged the work a bit too harshly on my first pass. Although the changes made by the authors are not huge, together they comprise a very effective system for reconstruction/view synthesis of objects with strong view dependent appearance (supported by strong qualitative/quantitative results in the paper, and additional results provided in the rebuttal).
> >
> > I would still suggest that the authors focus on improving clarity (e.g. by incorporating their response to reviewer 8TqS about section 3 into the paper), and also slightly tone down their claims. While I appreciate the distinction between specular highlights and reflections, specular highlights are a subset of effects caused by strong reflections -- so I would say that GaussianShader/NeRFCasting does attempt to model such affects, although they do not *specifically* focus on these effects.

---

> > > ### Author Response · Authors · 2024-08-12
> > >
> > > We thank the reviewer for the valuable comments and are glad to hear that our previous answers helped you better understand our work. We are keen to follow up on the provided suggestions:
> > >
> > > - We will incorporate our response to reviewer `8TqS` regarding section 3 into the paper to enhance clarity.
> > > - We will slightly tone down our claims and retain only the statement: "An anisotropic dataset has also been created to assess the capability of our model in representing anisotropy."

---

> ### Comment · Area_Chair_tLag · 2024-08-10
> **Please review author rebuttal!**
>
> Dear Reviewer,
>
> I wanted to gently remind you to please review the rebuttal provided by the authors. Your feedback is invaluable to the decision-making process, and if you feel that the rebuttal addresses any of your concerns, please consider updating your score accordingly.
>
> Thank you for your continued dedication to ensuring a fair and thorough review process!
>
> Best, Your AC

---

### Author Rebuttal · Authors · 2024-07-31

We thank all the reviewers for their valuable comments. We are glad and appreciate that the reviewers recognize that our proposed ASG appearance field and coarse-to-fine training are sound, efficient, and show significant performance improvements. We will polish our paper further and release our codes.



We would first like to clarify the contributions of Spec-Gaussian and the different variants presented in our paper. Following that, we will address the specific questions posed by each reviewer.



**Contribution and differences with GaussianShader:**

Reviewers `anMP` and `Ye8M` have raised concerns regarding our claim that “this work is the first to address specular highlights modeling in GS,” suggesting that prior works like GaussianShader have also attempted to tackle this issue. We respectfully disagree with this. Although both GaussianShader and our approach aim to handle view-dependent appearances, we fundamentally differ in how we model the underlying factors that cause these appearances:

- GaussianShader, GS-DR, and NeRF-casting mainly focus on reflective scenes, referring to the phenomenon where glossy objects reflect their **surrounding objects in the environment**. These methods primarily incorporate the objects’ geometry and environment map with a rendering equation to model view-dependent appearances caused by reflective surfaces.
- In contrast, our approach and Spec-NeRF focus on scenes with specular highlights, which are the bright spots of light that appear on shiny surfaces when viewed from a specific direction. These highlights result from the interaction between the **intensity of light sources and the material properties** of the object and are independent of other objects in the environment. Our method effectively captures specular highlights, as demonstrated in our paper (see Fig. 1 and Fig. 4), while previous methods based on Gaussian splatting have indeed been unable to model sharp specular highlights.

In sum, the efforts made by GaussianShader and our method are complementary and could be combined in the future to model view-dependent appearances in complex scenes and objects. This can be an area for future research. We will emphasize these differences in our paper to avoid any confusion. If the reviewers still find our claim inappropriate, we are very open to toning down this statement. Reviewers can see the illustration of specular highlights and reflection in the submitted rebuttal PDF.



**Explanation of different variants:**

- `Ours`, a method based on 3D-GS, referred to as the performance version, has $\tau_g=0.0005$.
- `Ours-light`, also based on 3D-GS, is called the light-version, with $\tau_g=0.0006$.
- `Ours-w/ anchor`, based on the Scaffold-GS, is referred to as the mini-version, with $\tau_g=0.0006$.

---

### Decision · Program_Chairs · 2024-09-25

**Decision:**

Accept (poster)

**Comment:**

This paper received positive scores of weak accept, weak accept, borderline accept and borderline accept. The AC concurs with the reviewers and recommends acceptance. The authors have promised changes in their rebuttal:

- Slightly tone down their claims wrt being the "first to address specular highlights modeling in GS."
- Incorporate suggestions from reviewer 8TqS regarding section 3, as well as address additional questions from 8TqS re: problems with the text
- Incorporate additional experiments provided in the rebuttal

The AC urges the authors to uphold their promises.